# The Role of PARPs in Inflammation—And Metabolic—Related Diseases: Molecular Mechanisms and Beyond

**DOI:** 10.3390/cells8091047

**Published:** 2019-09-06

**Authors:** Yueshuang Ke, Chenxin Wang, Jiaqi Zhang, Xiyue Zhong, Ruoxi Wang, Xianlu Zeng, Xueqing Ba

**Affiliations:** The Key Laboratory of Molecular Epigenetics of the Ministry of Education, Institute of Genetics and Cytology, Northeast Normal University, Changchun 130024, China

**Keywords:** poly(ADP)-ribosylation, PARP1, cancer, inflammation, metabolism, therapeutic approaches

## Abstract

Poly(ADP-ribosyl)ation (PARylation) is an essential post-translational modification catalyzed by poly(ADP-ribose) polymerase (PARP) enzymes. Poly(ADP-ribose) polymerase 1 (PARP1) is a well-characterized member of the PARP family. PARP1 plays a crucial role in multiple biological processes and PARP1 activation contributes to the development of various inflammatory and malignant disorders, including lung inflammatory disorders, cardiovascular disease, ovarian cancer, breast cancer, and diabetes. In this review, we will focus on the role and molecular mechanisms of PARPs enzymes in inflammation- and metabolic-related diseases. Specifically, we discuss the molecular mechanisms and signaling pathways that PARP1 is associated with in the regulation of pathogenesis. Recently, increasing evidence suggests that PARP inhibition is a promising strategy for intervention of some diseases. Thus, our in-depth understanding of the mechanism of how PARPs are activated and how their signaling downstream effecters can provide more potential therapeutic targets for the treatment of the related diseases in the future is crucial.

## 1. Introduction

Poly(ADP-ribose) polymerases (PARPs), also known as ADP-ribosyltransferases (ARTs), are a family of proteins that play key roles in various biological processes [1,2]. The PARP superfamily catalyzes either mono-ADP-ribose (MAR) or poly ADP-ribose (PAR) to target proteins using nicotinamide adenine dinucleotide (NAD^+^) as a donor; this process is also termed MARylation or PARylation, respectively [2,3]. PARylation is a reversible process in which covalently linked PAR can be hydrolyzed to free PAR by PAR glycohydrolase (PARG) or PAR hydrolase (ARH3) [4]. In addition to covalent PARylating proteins on classical amino acids such as glutamic acid, aspartic acid, and lysine, new amino acid sites such as arginine, cysteines, tyrosine and serine have been identified based on new experimental methods and techniques [5,6,7,8,9,10,11]. Notably, besides covalently PARylating target proteins, binding of proteins with PAR in noncovalent manner has also been reported to participate in a variety of biological processes [12,13].

Humans express at least 17 PARP enzymes that play important roles in regulating fundamental physiological processes in cells. Except for a splice variant of PARP13, PARP13.2, all other PARP members contain a classical PARP domain, which catalyzes ADP-ribosylation of target proteins (Figure 1). Based on their domain schema, PARP superfamily members can be divided into four sub-families: (1) DNA-dependent PARPs, which are activated by discontinuous DNA structures via amino-terminal DNA-binding domains. This sub-family includes PARP1, PARP2, and PARP3; (2) the tankyrase sub-family, including PARP5a (tankyrase1) and PARP5b (tankyrase2), contains five ankyrin (ANK) repeats, which serve as a binding platform that mediates protein-protein interactions; (3) CCCH (Cys-Cys-Cys-His zinc fingers) PARPs include PARP7 (also known as tiPARP), PARP12, PARP13.1, and PARP13.2. The CCCH domain mediates PARP binding to ribonucleic acid (RNA); (4) the macroPARPs contain macrodomain folds, including PARP9 (BAL1), PARP14 (BAL2 and CoaSt6), and PARP15 (BAL3), which can bind to ADP and derivatives [14] (Figure 1). As Figure 1 showed, PARP1, PARP2, and PARP5 catalyze PAR attaching to target, while other PARP members (PARP 3, 4, 6–8, and 10–16) catalyze MAR to target proteins. Two members of this family, PARP 9 and 13 appear to lack any enzymatic activity [3,4,14]. Therefore, based on their different sequences, it can be predicted that the PARP family is functionally diverse. This diversity of functions is reflected in different subcellular distributions, expression patterns, and diverse biological functions.

PARP1 is the prototypical and founding member of the PARP family, which can be divided into three functional domains: (1) the amino-terminal DNA-binding domain l (DBD) contains three zinc fingers, a nuclear localization signal (NLS), and a caspase-3 cleavage site; (2) the central automodification domain (AMD) contains numerous glutamic acid and aspartic acid residues, which is consistent with the fact that it is the main site of PARP1 automodification. This domain also contains a breast cancer susceptibility gene1 C-terminus (BRCT) motif, a protein-protein interaction motif, which is commonly found in proteins involved in DNA repair and control of the cell cycle [15], the PARP1 BRCT domain is required for the interaction between PARP1 with several partner proteins, including X-ray repair cross complement gene 1(XRCC1) [16] and histones [17]. PARylation of the AMD domain is thought to down-regulate PARP1 activity, due to the charge repulsion that develops between automodified PARP1 and DNA; (3) The C-terminal catalytic domain (CD) is the most well-conserved domain across the PARP family, it consists of a tryptophan-glycine–arginine-rich (WGR) motif and an “PARP signature” motif. The WGR motif is defined by conserved Trp, Gly, and Arg residues. The function of this domain remains uncertain, although it is speculated to be a possible nucleic acid binding region [18]. The “PARP signature” motif executes the catalytic function of PARP1, synthesizing PAR by using NAD^+^ as a substrate.

PARP1 is usually activated by DNA damage, and the modified target is primarily PARP1 itself, which is called PARP1 automodification. Furthermore, binding to non-damaged DNA structures is considered to be a valid determinant of PARP1 activation in the absence of abundant DNA fragmentation [19]. In addition, besides direct binding to DNA, PARP1 activity can also be stimulated by interactions with protein binding partners. For example, the DNA-binding transcription factor Yin Yang 1 (YY1) could bind to the BRCT motif in the automodification domain of PARP1, then stimulate PARP1 enzymatic activity [20]. In addition, studies have shown that post-translational modifications such as phosphorylation or acetylation, etc. are also alternative mechanisms of PARP1 activity [21]. Recently, our study showed that in response to lipopolysaccharide (LPS) or TNF-α exposure, the non-receptor tyrosine kinase c-Abl phosphorylates PARP1 at the conserved Y829 site. Tyrosine phosphorylation of PARP1 is required for PARP1 catalytic activity and NF-κB-dependent pro-inflammatory gene regulation [22]. Actually, a large body of data has now shown that PARP1 functions as a cellular rheostat, promoting different cellular response upon a wide range of type, duration, and strength of stress signals [4]. As the strength of stress stimulus increases, the levels of PARP1 activity and PAR synthesis increase, leading to different cellular outcomes. Weakly activity of PARP1 caused by LPS or TNFα can help cells response inflammation [22,23,24]; Mild or moderate stresses leads to transcription and DNA repair responses that help to maintain genome stability, in contrast, excessive activity of PARP1 resulted the depletion of NAD^+^ and accumulation of PAR, leading some metabolic disorder or cell death [4]. Activated PARP1 accounts for 85%-90% of the total PARP activity in cells, and PARP1 is the most extensively and intensively studied member of the PARP family [3,14,25]. As such, this article will focus on the important roles and molecular mechanisms of PARP1 in inflammation- and metabolic-related diseases.

## 2. PARP1 is Implicated in Disease via Altering Gene Expression

Studies have shown that PARP1 is rapidly activated under pathophysiological conditions and its activation is prolonged and sustained [26,27]. Indeed, over the past decade, a growing body of literature has revealed that PARP1 may play an important role by regulating the expression of disease-related genes (e.g., chemokines, pro-inflammatory mediators, and metabolic related factors) [28,29,30,31]. PARP1 may control gene expression by: (1) chromatin modulation; (2) transcriptional regulation; and (3) RNA regulation. In the following sections, we will discuss how PARP1 participates in human disease by altering gene expression.

### 2.1. Modulation of Chromatin

Numerous reviews have comprehensively summarized the ability of PARP1 to automodify itself or PARylate histones, and other chromatin-associated proteins to alter the chromatin structure, thereby leading to the fine-tuning of gene expression in situations of genotoxic stress signaling [2,32,33,34,35]. The linker histone, H1, has been identified as a major acceptor of PARylation [33,36]. Moreover, PARP1 and H1 have been found to compete for binding to target gene promoters, which contribute to the dynamic regulation of gene expression [37,38]. For example, activated PARP1 can PARylate H1 and cause an H1 displacement from chromatin, which makes the chromatin more open and amenable for the remodeling of progesterone-responsive regions under hormone treatment [39]. Moreover, NSD2, a histone methyltransferase, dimethylates histone H3 at lysine 36 (H3K36me2), which contributes to gene transcription activation. A very recent study showed that the PARylation of NSD2 by PARP1 decreases NSD2 histone methyltransferase activity and its association with chromatin in response to a DNA damaging stimulus [40].

Moreover, PARP1 has been reported to be a regulator of the function of insulators in the three-dimensional structure [31,41,42,43]. The CCCTC binding factor (CTCF), a chromatin insulator binding protein, is a major regulator of transcriptional coordination in three dimensions. PARylation of CTCF can influence the ability of CTCF to bind to the genome. Studies have shown that PARP1 and CTCF colocalize at specific sites throughout Epstein–Barr virus (EBV) genome, and PARP1 can cooperate with CTCF to regulate the latency of EBV [43]. EBV belongs to the family of gammaherpes viruses, and while a latent EBV infection is typically asymptomatic in healthy individuals, EBV is also associated with epithelial and several other types of lymphoid tumors [44]. During EBV latency, PARP1, and CTCF cooperate at specific sites. The PARylation of CTCF catalyzed by PARP1 promotes an open chromatin structure and active transcription by maintaining decondensed nucleosomes, H3K4me3, and preventing DNA methylation. Moreover, the PARP inhibitor, olaparib, eliminates the recruitment of CTCF to the viral genome, which leads to a decrease in EBV latent-related transcription [43,45,46]. Thus, PARP1 activity can be a regulator of the function of insulators in the three-dimensional structure, and the development of PARP inhibitors also provides new targets of EBV infection (Figure 2a).

### 2.2. Transcriptional Regulation

In addition, PARP1 can regulate gene expression by directly binding to promoters. Nitric oxide (NO) participates in a variety of physiologic and pathophysiologic processes in a diverse range of tissues, including the regulation of vascular tone and regional blood flow [47,48]. PARP1 has been reported to bind to the inducible nitric oxide synthase (iNOS) promoter to enhance nitric oxide (NO) production. Since excessive NO production can be cytotoxic to host cells, feedback inhibition of iNOS transcription represents a means of cellular protection. By s-nitrosylating the trans-activator PARP1, NO feedback inhibits iNOS gene transcription, decreasing PARP1 binding and/or action at the iNOS promoter [49]. Furthermore, a recent study showed that in apolipoprotein E-deficient mouse and aortic endothelial cells, the inhibition or knockdown of PARP1 could increase NO production and attenuate aortic atherogenesis through the down-regulation of arginase II [50] (Figure 2b). Additionally, a recent study has shown that the combination of NO donors and PARP inhibitors can be used as a new method of increasing cellular sensitivity to ionizing radiation [51].

Melanoma is the most aggressive form of skin cancer and is highly resistant to conventional chemotherapy. The CXC ligand 1 (CXCL1) chemokine produced by melanoma cells is a major effector of tumor growth. In normal melanocytes, inactive PARP1 attaches to the CXCL1 promoter and prevents NF-κB from binding to the promoter; however, in cancer cells, PARP1 activation and PAR modification have resulted in a loss of its binding to the CXCL1 promoter, thus allowing NF-κB binding and enhanced CXCL1 expression [52] (Figure 2b). This mechanism of PARP1 transcriptional regulation of CXCL1 will provide key information for developing strategies that can block the constitutive expression of this or other chemokines in the treatment of melanoma.

A major function of PARP1 in the regulation of the inflammatory process is to act as a transcriptional coactivator/corepressor [30,53]. In addition to the classical transcriptional factors (e.g., NF-κB or p53) [54,55,56,57], PARP1 also modulates the inflammatory response through regulating other important immune-related transcription factors. Some transcription factors (e.g., signal transducer and activator of transcription-6 [STAT-6] and activator protein 1 [AP-1]) are involved in the pathogenesis of asthma. STAT-6 is involved in the development of the Th2 cell-mediated immune response, mucus production, and allergic reactions. PARP1 enzymatic activity appears to be required for STAT-6 integrity. The level of STAT-6 mRNA expression was not altered in the spleens of either wild type (WT) or *Parp*^−/−^ mice subjected to ovalbumin challenge; however, a PARP1 gene deletion promotes STAT-6 protein degradation in the spleen following allergen stimulation. The down-regulation of STAT-6 coincided with a reduction in GATA-binding protein-3 and occupancy of its binding site on the IL-5 gene promoter; IL-5 is important for antigen-induced airway hyper-reactivity and pulmonary eosinophilia [58]. One study demonstrated a novel function of PARP1 in the lungs of allergen-exposed animals by influencing IL-5 expression. AP-1 is a transcription factor responsible for cytokine production and T helper cell differentiation in murine fibroblasts following oxidative challenge [59]. Kiefmann et al. showed that PARP1 contributes to acute lung injury via the up-regulation of iNOS through the activation of AP-1, but not NF-κB, during endotoxemia [60] (Figure 2c).

Additionally, PARP1 has a pro-inflammatory function in activated T cells. The interaction between nuclear factor of activated T cells (NFAT) and PARP1 facilitates IL-2 transcription [61]. In inactive T cells, NFAT is heavily phosphorylated and resides in the cytosol, and calcineurin-mediated dephosphorylation-regulated NFAT is shuttled to the nucleus. In the nucleus, NFAT interacts with coregulators (e.g., Fos-Jun, C/EBPs, and Fox3p), which form a composite transcription complex to regulate NFAT-mediated gene transcription. It has been reported that PARP1 binds to NFAT and PARylated NFAT at the E408 site located in the DNA-binding domain and increases its DNA-binding function (Figure 2c).

Transcriptional factor high mobility group box-1 protein (HMGB1) is a key late mediator of sepsis and is now known to be associated with various types of inflammation [62]. Under DNA damage stimulation, activated PARP1 was demonstrated to increase the level of HMGB1 accumulation in the cytosol, which establishes the ability of cells to release this potent inflammatory mediator upon subsequent necrotic death [63]. The translocation of HMGB1 from the nucleus to the cytoplasm is dependent on the modification of HMGB1 itself. Its reported acetylation of HMGB1 is necessary for HMGB1 shuttling and exported acetyl-HMGB1 cannot return to the nucleus. Studies showed that PARP1 catalyzed PARylation of HMGB1, which facilitated acetylation of HMGB1. The crosstalk between PARylation and acetylation directed the migration of HMGB1 into the cytoplasm as well as release into the extracellular space [64,65]. Furthermore, C/EBPβ is a key pro-adipogenic transcription factor that has been found to be PARylated by PARP1. PARylated C/EBPβ inhibits its DNA-binding and transcriptional activities, preventing differentiation in the absence of an adipogenic signal. During differentiation, cells with reduced nuclear NAD^+^ concentrations lead to low PARP1 activity, which allows C/EBPβ to initiate the adipogenic transcription program [66,67] (Figure 2c).

Interestingly, PAR generated by PARP1 and PARG also been reported participate in transcriptional regulation. For example, p53 displays a noncovalent high-affinity interaction with PAR. Noncovalent bind by free PAR and covalent PARylation of p53 orchestrates each other, affected p53-dependent DNA-binding properties, transcriptional function under genotoxic stress condition [56]. Another example showed that the nuclear free PAR could be further broken down by pyrophosphatase NUDIX5 to produce adenosine triphosphate (ATP) in the presence of pyrophosphate. The ATP then participated in the ATP-dependent chromatin remodeling and transcriptional programmes in response to hormone-dependent signaling [68].

### 2.3. Regulation of RNA Metabolism

Extensive studies have documented that transcriptional activation constitutes the primary mode by which PARP1 modulates gene expression. Recent reports have demonstrated that several RNA-binding proteins (RBPs) are modified by PARylation to regulate post-transcriptional processes [24,69]. These studies have been fruitful and have revealed new aspects of the PARP biology in diseases via the influence of mRNA metabolism.

Fragile X associated tremor ataxia syndrome is a neurodegenerative disorder caused by the inhibition of fragile X mental retardation (FMR1) gene expression. It is believed that the presence of the triplet (CGG) repeats in the 5’UTR of the FMR1 gene leads to low FMR1 gene expression [70]. It has also been shown that mouse heterogeneous nuclear ribonucleoprotein (hnRNP) A2/B1 binds directly to the CGG repeats of the FMR1 gene and suppresses fragile X CGG premutation repeat-induced neurodegeneration [71]. PAR generated by PARP1 has been reported to influence hnRNP binding to target mRNA, and the noncovalent binding of hnRNA to PAR promotes the dissociation of hnRNP from RNA. Although the study did not define PARP1 function, PAR binding to hnRNPs causes the dissociation of hnRNPs from RNA, which suggests a reduction of hnRNP noncovalent binding with PAR, via the administration of a PARP inhibitor, can increase the amount of unmodified hnRNPs and may improve treatment for some neurodegenerative diseases [69,72] (Figure 2d).

Moreover, hnRNPA1 is believed to correlate with cancer cell proliferation in a variety of tumors, including lung cancer, colorectal cancer, and gliomas [73,74,75]. The pyruvate kinase isoform 2 (PKM2) could promote the Warburg effect by interacting with the subunit of hypoxia-inducible factor 1 and induce the expression of hypoxia response genes during tumorigenesis, whereas the other isoform, PKM1, promotes oxidative phosphorylation [75]. These two isoforms result from mutually exclusive alternative splicing of the PKM pre-mRNA, depending on the inclusion of either exon 9 (PKM1) or exon 10 (PKM2). HnRNPA1 and A2 bind to an intronic splicing silencer of PKM1, repressive to sequences flanking exon 9, resulting in exon 10 inclusion, which ensures a high PKM2/PKM1 ratio in tumor cells [75]. Because the binding of PAR to hnRNP inhibit the RNA-binding activity of hnRNP A1, PARP inhibitors may have a therapeutic benefit for some hnRNP A1-dependent tumors by interfering with cancer cell metabolism (Figure 2d).

Another important RNA-binding protein, HuR, has been shown to interact with PARP1 and is PARylated by PARP1 [24]. HuR binds to a series of mRNAs in cells and regulates mRNA stability, including cytokines/chemokines and pro-inflammatory factors. Chemokines are small proteins that direct circulating leukocytes to the site of inflammation or injury. Although originally studied based on their role in inflammation, it is now known that chemokines and their receptors play a key role in guiding the movement of monocytes, producing adaptive immune responses, and contributing to the pathogenesis of several diseases [76,77,78,79,80,81]. The mRNA stability of a series of pro-inflammatory cytokines/chemokines (e.g., *Cxcl1*, *Cxcl2*, *Cxcl13*, and *Il-1β*) during LPS-induced pro-inflammatory signaling in RAW 264.7 macrophages is influenced by PARP1-HuR signaling, and the PARP inhibitor, PJ34, can significantly inhibit LPS-induced inflammation in the mouse lung [24]. Our previous studies have shown that the binding of HuR and its target gene mRNA are influenced by the suppression of PARP1 activation. These results suggest that in some forms of acute inflammation, PARP1 is involved in the regulation of pro-inflammatory gene mRNA stability by PARylatinged HuR [31] (Figure 2e). In addition, HuR can bind to PARG mRNA under the stimulation of PARP inhibitors, and the inhibition of HuR binding to PARG mRNA helps to improve PARP inhibition therapy in pancreatic cancer cells [82].

In addition to its diverse roles in differentiation, response to damaging stimuli, and the inflammatory response, HuR has also been reported to be associated with tumorigenesis. HuR can promote the expression of proteins that increase proliferation, enhance cell survival, reduce apoptosis, improve angiogenesis, reduce immune recognition, and facilitate invasion and metastasis that are considered the hallmarks of cancer [83]. In normal tissues, HuR is predominantly located in the nucleus, whereas in several cancer cases (e.g., small cell lung cancer, colon cancer, gastric adenocarcinoma, ovarian cancer, and glioma disease), the cytoplasmic localization of HuR is elevated. This suggests a significant role for HuR following cancer development, as the shuttling of HuR from the nucleus to the cytoplasm favors cancer development [84,85,86]. The PARylation of HuR by PARP1 was found to enhance HuR nucleoplasmic shuttling and mRNA-binding, and promote the mRNA stability of target transcripts. Although it is unclear how PARP1 and PARylation affect the translocation of HuR from the nucleus to the cytoplasm, PARP inhibition by olaparib or PJ34 significantly reduced the accumulation of HuR in the cytoplasm. This indicates that PARP inhibition is a potential strategy for treating cancers that are closely related to HuR translocation by inhibiting HuR shuttling (Figure 2e).

## 3. Insight into the Involvement of PARP1 in the DNA Damage Response and Metabolic Diseases

The human genome is constantly being attacked by various endogenous or exogenous stresses, including oxidative stress, telomere erosion, oncogenic mutations, and genotoxic stress [87]. The accumulation of DNA damage is involved in the regulation of tissue inflammation, metabolic homeostasis, and age-related diseases [88]. Once DNA is damaged, cells will immediately initiate the pathways to repair the DNA damage, including base excision repair (BER), nucleotide excision repair (NER), homologous recombination (HR), and the non-homologous end-joining (NHEJ). Single-strand breaks are repaired by BER or NER, whereas double-strand DNA breaks are repaired by NHEJ or HR [87]. PARP1 activation via DNA damage is believed to play an important role in many diseases, including breast cancer and diabetic neuropathy [88].

Our understanding of the mechanism of PARP1 in DNA damage repair provides novel insight into the treatment of some DNA damage-related diseases or tumors in the future. In mammalian cells, both DNA single-strand breaks (SSBs) and DNA double-strand breaks (DSBs) trigger PARP1 to synthesize long and branched PAR chains at DNA damage sites to recruit DNA repair factors. Once a DNA single-strand break occurs, the DBD of PARP1 recognizes the DNA break site and is then activated. PARP1 transfers PAR to itself and recruits other target proteins, including histones, topoisomerases, DNA helicases, and single-strand break repair; this facilitates the relaxation of the chromatin superstructure, protein-protein interactions, and the DNA-binding ability of the members of the DNA repair machinery (Figure 3a) [89]. Mammalian cells employ at least two sub-pathways of non-homologous end-joining for the repair of DNA double-strand breaks. One is the canonical DNA-PK complex (Ku70/Ku80 and DNA-PKcs)-dependent form of non-homologous end-joining (D-NHEJ); another is an alternative, slowly operating, error-prone backup pathway (B-NHEJ) [90,91]. The role of PARP1 in double-strand break repair (DSBR) is complex. In the HR and PARP1-dependent B-NHEJ pathway, PARP1 is thought to recognize and bind to the site of DSB and is subsequently activated. Then the activated PARP1 rapidly recruits the DNA damage sensors meiotic recombination 11 (Mre11) and Nijmegen breakage syndrome protein 1 (Nbs1) to sites of DSBs, Mre11 has a putative PAR-binding domain, suggesting that its recruitment could be dependent on the activity of PARP1. Mre11 protein interacts with the Rad50 recombinase to form MRN (Mre11-Rad50-Nbs1) complex, after which the MRN complex initiates the repair process [88,92] (Figure 3b, left). In eukaryotic cells, DSB repair is typically carried out by non-homologous end-joining (NHEJ) and D-NHEJ is the major NHEJ pathway in vivo, for which initiation requires the DNA-PK complex [93,94]. Whether PARP1 directly participates in D-NHEJ remains controversial (Figure 3b, right). PARP1 is overexpressed in a variety of cancers, including glioblastoma, prostate and breast cancer [95,96,97,98,99,100]. Up-regulated PARP1 enhances the anti-apoptotic properties of tumors, leading to resistance to DNA damage therapeutics. Cancer cells use PARP1 to initiate a variety of DNA repair pathways, particularly BER and HR, to prevent cell death caused by the accumulation of cytotoxic DNA damage, making it a good candidate for sensitizing cancer cells to the cytotoxic effects of DNA damaging agents [101,102,103].

Recently, a list of studies has shown that Ser ADP-ribosylation (S-ADPr) is involved in genome stability regulation and the DNA damage response (DDR) processes. This modification is highly dependent on histone PARylation factor 1 (HPF1). Besides PARP1 itself, S-ADPr has been found as a widespread post-translational modification that targets hundreds of proteins, such as histones and high-mobility group (HMG), those proteins have important roles in chromatin regulation, DNA repair, and cancer progression, which imply the significant of S-ADPr in the regulation of DNA damage response and the maintenance of genome stability [5,6,8]. The discovery of S-ADPr have added considerable depth to our understanding of the function and versatility of ADPr signaling in the cell, and the potential physiological and pathological significance needs further study.

In addition, the accumulation of DNA damage is also involved in the regulation of metabolic homeostasis through influences on cellular metabolism and the endocrine system [104]. Similar to the PARP family, sirtuin (SIRT) proteins are also an NAD^+^ dependent enzyme that play an important metabolic role in the cell. Observations indicate that the decline of NAD^+^ consumption by PARP1 activity correlates with a down-regulation in SIRT activity [104]. PARP1 can modulate SIRT1 transcriptional activity by depleting the pool of NAD^+^ co-factors necessary for SIRT1 deacetylase activity. Among the SIRT family, SIRT1 is the most well-studied. SIRT1 may play a crucial role in metabolic homeostasis by regulating the activity of several transcriptional regulators, including peroxisome proliferator-activated receptors (PPARs), sterol regulatory element binding protein (SREBP), and cAMP response element binding protein (CREB) [105,106]. The activity of SIRT1 has been reported to protect against aging-related and neurodegenerative diseases. Moreover, the function of SIRT1 has also been shown to be strongly associated with cancer and apoptosis [107]. Given that SIRT1 and PARP1 are both NAD^+^ dependent, these enzymes compete for the NAD^+^ pool. Once DNA is damaged, PARP activation depletes cellular NAD^+^ pools, hampering cellular energy metabolism by reducing SIRT1 activity [30] (Figure 4).

## 4. PARP-Dependent Death and Related Diseases

Mild genomic stress induces PARP1 activation to repair damaged DNA and maintain genomic stability (Figure 5a). Meanwhile, excessive PARylation occurs from massive DNA disruption (e.g., traumatic brain injury and ischemia/reperfusion injury) in various organs induces cell death. PARP1-dependent cell death is caspase-independent and termed *parthanatos*. *Parthanatos* has been implicated in various diseases, including stroke, Parkinson’s disease, heart attack, and diabetes [108,109,110,111,112,113] (Figure 5b). It has long been thought that the cell death caused by the excessive activation of PARP1 occurs through the catalytic consumption of NAD^+^ followed by a reduction in ATP and bioenergetic collapse. However, recent studies have shown that energy expenditure is not a determinant of cell death. Moreover, studies indicate that PAR content is a major cause of cell death [88,89]. PAR is synthesized in the nucleus but can be transferred from the nucleus to the cytoplasm. Cytoplasmic PAR can cause the nuclear transfer of apoptosis-inducing factor (AIF) from mitochondria, ultimately leading to cell death [114,115].

AIF is an important mitochondrial flavoprotein involved in mitochondrial energy-generating processes [110]. AIF has also been identified as a mitochondrial protein that is involved in cell death. In the nucleus, AIF binds to DNA to induce the peripheral chromatin condensation and high-molecular-weight DNA fragmentation that is a hallmark of cellular apoptotic programs [116]. Once PARP1 becomes activated, PAR generated by PARP1 is hydrolyzed by PARG into free PAR. In addition, AIF is a high-affinity PAR-binding protein. PAR binding by AIF is required for its release from the mitochondria, translocation to the nucleus, and the induction of cell death [110,111,115]. Recently, it was reported that macrophage migration inhibitory factor (MIF) is required for PARP1-dependent parthanatos. MIF is predominantly localized to the cytosol of cells, where AIF interacts with MIF and recruits MIF to the nucleus, inside which MIF cleaves genomic DNA into large fragments and induces cell death under conditions of oxidative stress [117] (Figure 6).

The concept of *parthanatos* is primarily focused on the study of neuronal cell death [118,119]. Parkinson’s disease (PD) is the second most common neurodegenerative disorder. During the pathogenesis of PD, monomeric a-synuclein assembles into higher ordered structures, and these α-syn preformed fibrils (PFFs) ultimately become pathologic and promote neuronal cell death. Recent studies have shown that α-syn PFF-induced neurotoxicity is PARP1-dependent. PAR generated by PARP1 causes the formation of a more toxic α-synuclein strain, resulting in accelerated pathologic a-synuclein transmission and toxicity. The inhibition or deletion of PARP1 reduces α-synuclein PFF-induced cell death [120].

Glutamate is the main excitatory neurotransmitter that regulates the normal physiological activities of the brain. Excess glutamate acts on N-methyl-d-aspartate (NMDA) receptors to induce many neurological diseases, including trauma and stroke. Glutamate excitotoxicity is mainly mediated through the influx of calcium through the NMDA receptor, leading to PARP1 activation and PAR polymer production. Iduna, an NMDA-induced neuroprotective protein, has been reported to protects against stroke induced neuronal injury. Iduna contains a PAR-binding motif (PBM) consists approximately 20 amino acids, PBM domain is high affinity with PAR, by this way, Iduna competed with AIF for PAR binding, thereby blocking the translocation of AIF from mitochondria to the nucleus [121]. Moreover, Iduna also been found is a PAR-dependent E3 ligase that binds and ubiquitinates both PARylated and PAR-binding proteins via its PBM domain, marking these proteins for ubiquitin proteasomal degradation [96,122]. These studies suggest that the inhibition of PARP1 activation may be used as a strategy for the treatment of some neurological diseases.

## 5. Other PARPs in the Regulation of Human Diseases

The PARP superfamily contains 17 members. Over the past decade, studies have begun to reveal the physiological and pathological roles of the other PARP family members. Here, we will highlight some of the pathophysiological functions of other PARPs, as depicted in Figure 7.

The PARP2 sequence are primarily related to that of PARP1. PARP2 is a 62 kDa nuclear PARP member capable of automodification and interacts with several members of the base excision DNA repair machinery [123,124]. Similar to PARP1, PARP2 has also been recognized as a central component of the base excision repair/single-strand break repair process (BER/SSBR) [87,106]. Similar to *Parp1*
^−/−^ mice, *Parp2*^−/−^ mice also exhibit increased genomic instability and sensitivity to ionizing radiation and other DNA damaging agents [97,125,126]. Recent studies have shown that PARP2 binds to PARP1-neosynthetized PAR chains at DNA damage sites through its unstructured N-terminal region [127]. Although studies of PARP2 are limited compared to PARP1, PARP2 has also been shown to regulate some diseases. Moreover, PARP2 can function as a transcriptional regulator and participate in metabolic homeostasis [128]. PARP2 regulates adipocyte differentiation by acting as a positive regulator of the peroxisome proliferator-activated receptor γ (PPARγ) [129]. PPARγ is the main protein orchestrating the differentiation and function of white adipose tissue, which controls energy, lipid and glucose homeostasis [130]. The regulation of PPAR by PARP2 suggests the possibility of targeting PARP2 in the treatment of diseases (e.g., obesity and related metabolic disorders). Others have demonstrated that PARP2 regulates oxidative metabolism by functioning as a negative regulator of the SIRT1 promoter [128]. SIRT1 affects metabolism by deacetylating key transcriptional regulators of energy expenditure. Deletion of PARP2 in mice increases SIRT1 levels, which promotes energy expenditure [128]. Furthermore, *Parp2*^−/−^ mice have shown obvious protection against diet-induced obesity [128]. In addition, *Parp2*^−/−^ mice exhibit attenuated neuronal inflammation and lymphocyte infiltration in a mouse model of multiple sclerosis and were protected from focal cerebral ischemia [131]. These studies support the importance of PARP2 in various pathophysiological processes.

PARP3, as a mono-ADP-ribosylase, has also been reported play a critical role in DSB repair [132]. PARP3 can associate with some DNA repair factors, including DNA-PKcs, DNA ligase III, Ku70, and Ku80 [133]. In addition, PARP3 has also been shown to control the TGFβ- and ROS-driven epithelial-to-mesenchymal transition and stemness in human breast cancer cell lines [134]. PARP3 overexpression is associated with a poor prognosis in patients with breast cancer following chemotherapy and thus, PARP3 inhibitors may be a potential therapeutic strategy for the treatment of breast cancer [135,136].

PARP5a and PARP5b are multifunctional PARP family members that control a variety of cellular processes by interacting with target proteins and regulate their interactions or stability through PARylation (e.g., DNA damage repair, maintenance of telomere length, and mitosis), and participate in Wnt and Notch signal transduction [89,137,138,139]. In response to DSB, PARP5 binds to DNA damage checkpoint protein 1 (MDC1) and is recruited to DNA lesions. PARP5 also stabilizes the NHEJ protein, DNA-PK, and activates the G2/M checkpoint [140]. Furthermore, together with PARP12 and PARP13, PARP5a aggregates the protein complex response to cellular stress under both physiological and pathological conditions [141,142].

PARP6 is a new member of the PARP family that functions as a tumor suppressor by negatively regulating cellular proliferation [143]. PARP6 overexpression inhibits S-phase progression and cellular proliferation. While the catalytic domain of PARP6 is required for PARP6 function, PARP6 mutations lacking the catalytic domain had no effect on the cells [143]. PARP6 also plays a key role in suppressing colorectal cancer progression and its protein expression in human colorectal cancer was linked to a good prognosis [143,144,145]. On the other hand, new studies showed that PARP6 could maintain centrosome integrity via the direct MARylation of Chk1 and modulation of its activity in breast cancer cells. Treatment with AZ0108, a PARP6 inhibitor, led to apoptosis in a subset of breast cancer cells [146,147]. These studies suggested the dual role of PARP6 in different types of tumors.

PARP7, also known as 2,3,7,8-tetrachlorodibenzo-p-dioxin (TCDD)-inducible poly(ADP-ribose) polymerase (or TiPARP, ARTD14) is an uncharacterized member of the PARP family. Studies have shown that PARP7 is involved in the negative feedback regulation of the aryl hydrocarbon receptor (AHR) signaling pathway [148]. AHR is a member of the basic helix-loop-helix Period-AHR nuclear translocator (ARNT)-singled minded (bHLH-PAS) family and is a ligand-activated transcription factor. Moreover, AHR is well-known for its ability to mediate the toxic responses towards environmental contaminants (e.g., TCDD). In addition, AHR regulates immune infection, inflammation, and participates in cancer progression [149,150]. Furthermore, PARP7 positively regulates liver X receptor α (LXRα) and LXRβ activity. LXRs act as oxysterol receptors and are important physiological regulators of lipid, cholesterol, and glucose metabolism, as well as the inflammatory pathways [151]. The above findings demonstrate the importance of PARP7 as a co-activated transcription factor in the nucleus.

PARP10 has been found to suppress the activation of NF-κB by inhibiting the translocation of p65 into the cell nucleus and results in reduced target gene expression in HeLa and U2OS cells [152]. By contrast, PARP12 has been shown to colocalize and interact with members of the NF-κB signaling pathway. In addition, the overexpression of PARP12 activates NF-κB signaling, thus promoting IL-8 secretion in response to extracellular ligands [153,154]. Furthermore, PARP12 has recently been found to accumulate in cytoplasmic stress granules known to regulate mRNA translation and stability in response to stress [141].

The RNA-binding protein, PARP13, is another PARP member. Besides its role in targeting viral RNA degradation, PARP13 also helps to sensitize cells to TRAIL-mediated apoptosis by inhibiting the expression of the pro-survival receptor TRAILR4, thereby limiting cancer survival [155]. Indeed, multiple cancers exhibit low PARP13 expression compared to normal tissue, suggesting that PARP13 is both an important pro-inflammatory and proapoptotic agent [156].

Similarly, PARP14 also functions as a regulator of gene transcription. PARP14 is expressed in lymphoid organs and lymphocyte cell lines, plays an important role in signal transduction, and has been shown to be associated with asthma pathogenesis. Under non-IL-4 stimulating conditions, PARP14 acts as a transcriptional repressor by recruiting histone deacetylases 2 (HDAC2) and histone deacetylases 3 (HDAC3) to IL-4-responsive promoters. In the presence of IL-4, the catalytic activity of PARP14 facilitates the release of histone deacetylases (HDACs), which allows STAT-6 to bind to the promoter region of its target genes, thereby activating STAT-6-dependent transcription [157]. Interestingly, a recent study showed that PARP14 is also involved in the regulation of mRNA stability. PARP14 interacts with tristetraprolin, a dominant mRNA-destabilizing factor [158], and forms a complex that binds to the 3′UTR of tissue factor mRNA, thereby promoting its degradation [158]. Tissue factor is a transmembrane cell surface glycoprotein, which is a key mediator of thrombosis and inflammation, implicating that PARP14 is an important factor during mRNA decay in vascular inflammation.

Another member of the PARP family, PARP9, has been reported to play opposing roles to PARP14 in macrophage activation. The knockdown of PARP14 increased pro-inflammatory gene expression and STAT1 phosphorylation, whereas PARP9 silencing inhibited pro-inflammatory gene and STAT1 phosphorylation in IFNγ-treated cells. PARP14-induced MARylation of STAT1 was found to be inhibited by PARP9 [159].

Moreover, PARPs also regulate the unfolded protein response (UPR) of the endoplasmic reticulum (ER). PARP16 (also known as ARTD15) is a tail-anchored ER protein, which is inserted into the ER membrane. It appears that both PARP16 itself and its catalytic activity are required for the ER stress responses by regulating the UPR signaling pathway [160].

## 6. Therapeutic Approaches for Targeting PARPs

Most of the PARP inhibitors have been tested in clinical trials aimed at cancer therapy. In 2014, the PARP inhibitor, Olaparib (AstraZeneca’s Lynparza), was first approved by both the European Medicines Agency (EMA) and the U.S. Food and Drug Administration (FDA) as a cancer treatment targeting advanced ovarian cancers associated with defective BRCA1/2 genes. To date, the use of synthetic lethality has been most effectively exploited in the context of BRCA1/2-deficient breast and ovarian cancers [161,162,163]. Interestingly, recent studies also show the potential efficacy of PARP inhibition in sporadic tumors without DNA repair defects (e.g., regulating the expression of tumor-related genes and suppressing angiogenesis). Moreover, in many disease states beyond cancers, PARP inhibitors have been explored as potential therapeutics to prevent cell death, tissue damage, and dysfunction-associated pathologies (e.g., cardiovascular diseases, autoimmune and inflammatory diseases, neurodegenerative diseases, stroke, and diabetes) and any associated complications [29,164,165,166,167,168,169,170]. For example, small molecule phytoisoflavones have been widely used for the treatment of various diseases, including infectious and non-infectious inflammation [171,172]. Our previous study showed that daidzein is one type of phytoisoflavone that directly interacts with PARP1, inhibits p65 protein PARylation, and binds to PARP1, which is required for the transcriptional modulation of pro-inflammatory chemokines (e.g., Cxcl2); this implies that daidzein plays a role on suppressing pro-inflammatory gene transcription via decreasing PARP1 activity in TNFα-stimulated lung epithelial cells [173]. Also, we have shown that 17β estradiol and the hormone receptor ERα, they inhibit PARP activation, and increase cell viability. Estrogen significantly inhibits H_2_O_2_-induced PARP1 activation and nuclear fragmentation, which was abrogated when ERα was knocked down [174]. The results indicate that hormone therapy can interfere with diseases associated with PARP1 activation. In addition, vitamin D protects keratinocytes from PARP1 overactivation when cells are exposed to sunlight, suggesting that vitamin D has both pharmacological and anti-inflammatory effects [175].

Although the primary function of PARPs is to maintain genomic integrity, the overactivation of the family under an extensive and persistent DNA damaging environment promotes inflammatory conditions. In addition, excessive PARP activation is often observed in many diseases. PARPs are key players in the inflammatory response and the inhibition of PARPs limits the proliferation of many cancer cells [26,27,95,96,97,98,99,100]. Therefore, methods targeting PARPs have potential therapeutic applications in both cancer and inflammatory diseases [98,103,136,142,165,166,176,177,178]. Unfortunately, current research findings showed that in patients as well as in mouse models expressed PARP inhibitor resistance. This resistance is mainly due to overexpression of the drug efflux transporter gene; directly affects the activity of PARPs; leads to HR reactivation and affects the stability of the replication fork [179,180,181,182,183,184,185,186]. In addition, most of the current PARPs inhibitors have a relatively broad-spectrum for the PARP family; however, it has become increasingly evident that the PARP family members have very different functional roles in specific physiological processes (see above). Therefore, it should be realized that the development of combination treatments of PARP inhibitor with HR-inhibiting or other drugs and specific inhibitions for PARP family members may have pleiotropic effects and pave the way for novel treatment options to target resistance mechanisms.

## 7. Conclusions

In conclusion, PARP controls many of the genes involved in regulating immune responses, cell survival, and inflammation and metabolic disorders. Therefore, targeting PARP has potential therapeutic applications in several diseases associated with inflammation and metabolism.

## Figures and Tables

**Figure 1 cells-08-01047-f001:**
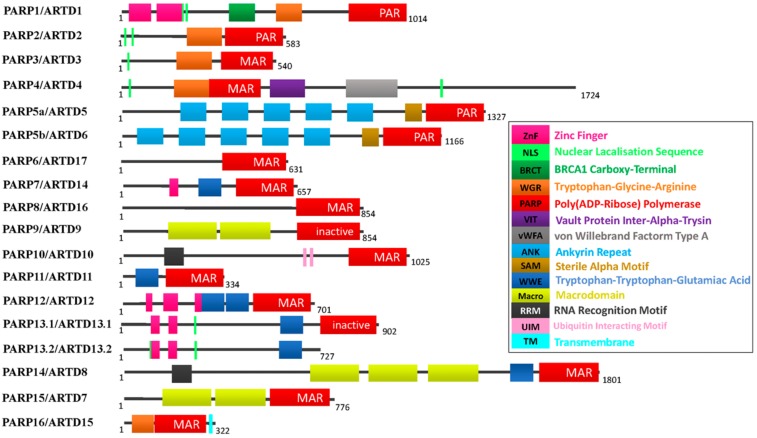
The PARP superfamily. Different color boxes indicate protein domains.

**Figure 2 cells-08-01047-f002:**
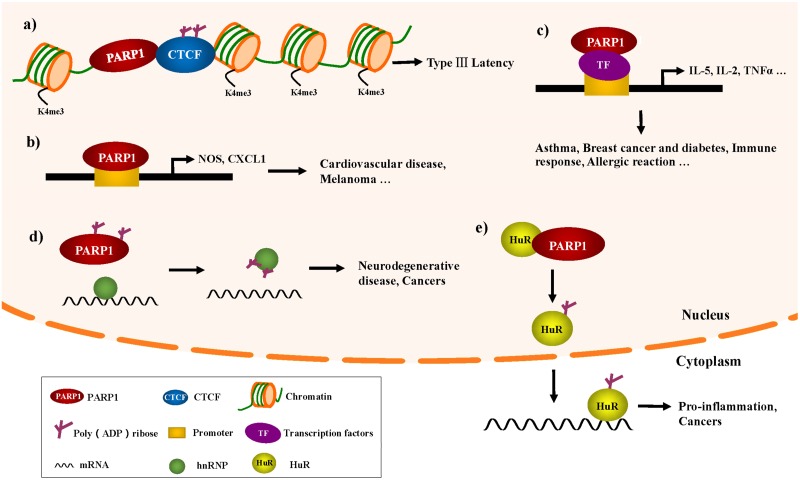
Significance of PARP1 in diseases by regulating gene expression. (**a**) In type III EBV latency, CTCF PARylation by PARP1 acts at the Cp promoter to maintain the open chromatin landscape and transcription; (**b**) PARP1 could participate in inflammation-related gene expression by the direct binding to NOS or CXCL1 promoters. NOS, nitric oxide synthase; (**c**) PARP1 acts as a transcriptional coactivator/co-repressor involved in the inflammatory process via the regulation of NF-κB, AP-1, and other transcription factors; (**d**) Noncovalent binding of hnRNA to PAR promotes the dissociation of hnRNP from RNA, suggesting that PARP inhibitors can improve the treatment of certain neurodegenerative diseases; (**e**) PARP1 PARylated HuR increases the accumulation of HuR in the cytoplasm, which leads to the functional activation of HuR, allowing it to bind to inflammation-related mRNAs.

**Figure 3 cells-08-01047-f003:**
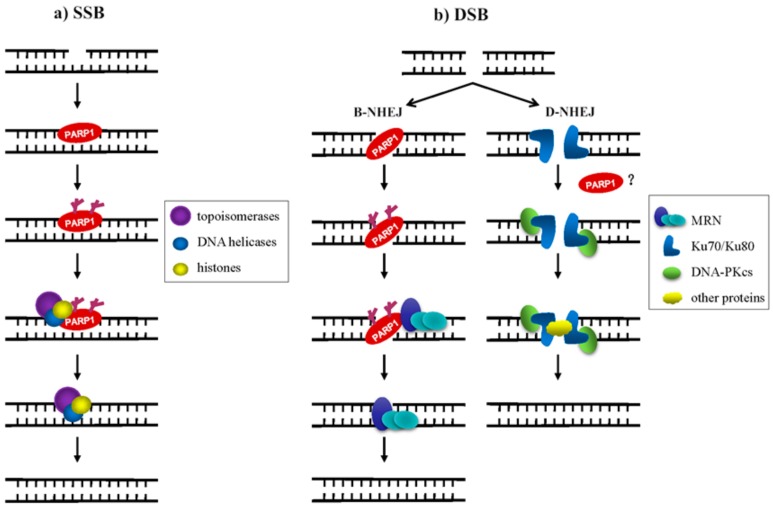
Role of PARP1 in DNA damage repair. (**a**) Ionizing radiation (IR) or reactive oxygen species (ROS) induces cell SSBs. PARP1 recognizes the DNA break site and recruits topoisomerase, DNA helicases, and histones to complete SSBR; (**b**) in the PARP1-dependent B-NHEJ pathway, PARP1 is thought to recognize and bind to the DSB site and once activated, recruits the MRN complex to DSB and completes the DNA repair (left). Whether PARP1 directly participates in D-NHEJ remains disputable (right).

**Figure 4 cells-08-01047-f004:**
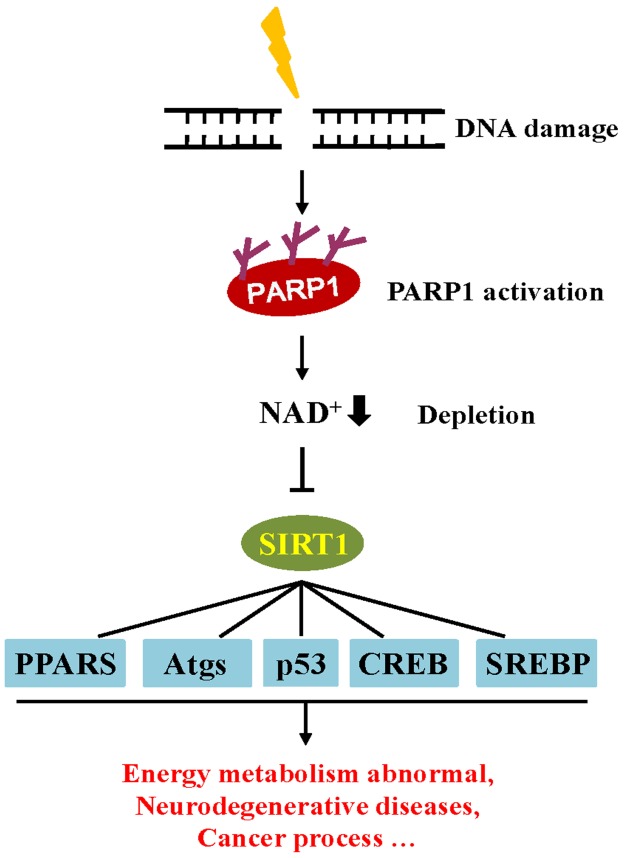
The crosstalk between PARP1 and SIRT1 in the DNA damage response. DNA damage activates PARP, leading to the depletion of NAD^+^ and induced inactivation of SIRT1 activity, which in turn, influence protein targets involved in metabolic regulation. PPARs, peroxisome proliferator-activated receptors; Atgs, autophagy-related proteins; SREBP, sterol regulatory element binding protein; CREB, cAMP response element binding protein.

**Figure 5 cells-08-01047-f005:**
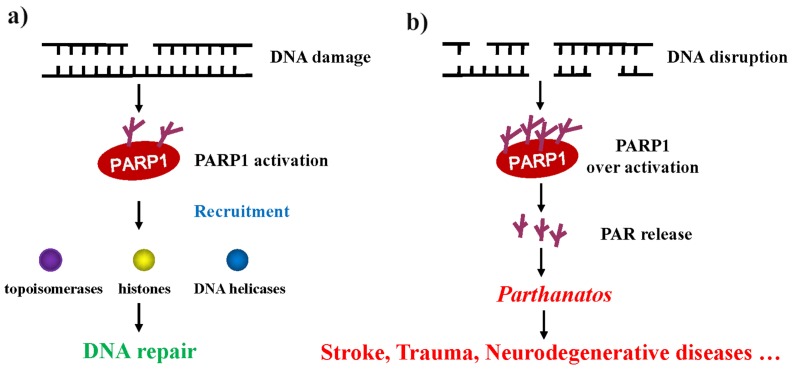
PARP-dependent cell death. (**a**) Low level of DNA damage breaks activate PARP1 to induce DNA repair via the recruitment of the DNA damage repair factor process; (**b**) when the DNA damage is beyond the threshold of repair, excessive PAR generated will cause cell death via the process of *parthanatos*.

**Figure 6 cells-08-01047-f006:**
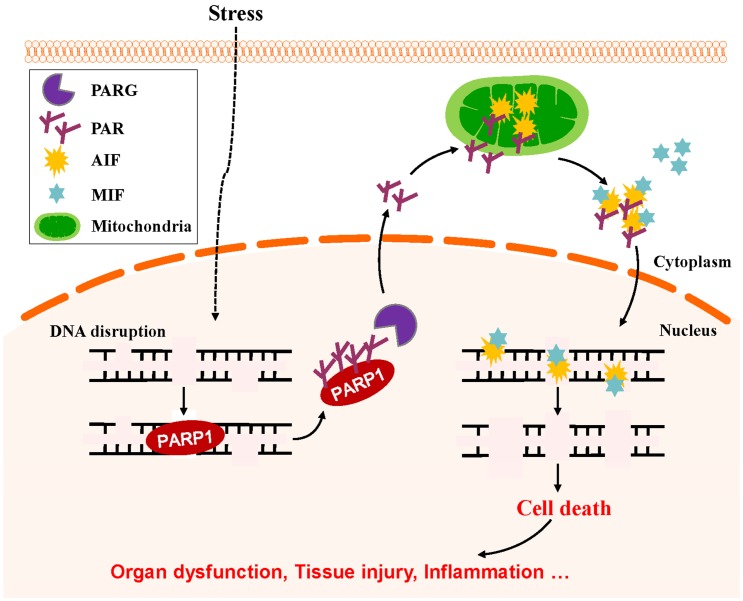
PARP1-mediated parthanatos by triggering AIF release from mitochondria. DNA damage in cell-stress conditions leads to PARP1 activation. PAR generated by PARP1 is hydrolyzed by PARG into free PAR and is translocated to the mitochondria, which induces AIF release from the mitochondria, translocation to the nucleus, and induction of the cell death.

**Figure 7 cells-08-01047-f007:**
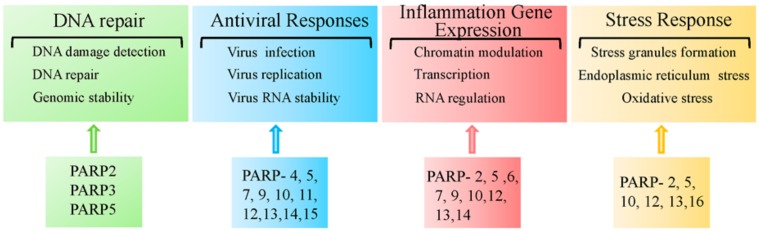
Other PARP functions in human diseases.

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
