# Peer review of "The Role of PARPs in Inflammation—And Metabolic—Related Diseases: Molecular Mechanisms and Beyond"

_cells, 2019, doi:10.3390/cells8091047_

Round 1
Reviewer 1 Report
Ke et al. provide an overview of PARP1 functions and on how some of them may relate to disease. The review focuses mostly on PARP1, other PARP family members are discussed in less detail, partly because they are less studied. Functional aspects that are covered include roles in regulation of chromatin, transcription and RNA metabolism, DNA repair, cellular metabolism, and PARP1-dependent cell death. The manuscript ends with a short paragraph on PARP inhibitors and their therapeutic potential. Due the growing interest in PARP inhibitors and PARP biology, many comprehensive reviews have already been published on the topic, and it may thus be difficult to find a new and original angle. Nevertheless, the authors could try to work more towards providing a synthesis of newly emerging concepts rather than listing individual and sometimes poorly connected research findings.
Specific points:
Given that the main focus is on PARP1, the title seems too broad. Along the same line, the manuscript only scratches the surface of the roles of PARPs and PARP inhibitors in different cancers, and it would therefore be more appropriate to choose a title more specific than “Human Diseases” and perhaps focus the manuscript accordingly, e.g. on inflammation and metabolic disorders. The manuscript would benefit from more clearly discriminating, whenever possible, between PARP functions that depend on covalent modification of target proteins with PAR versus functions that are associated with non-covalent interactions with PAR chains, especially if the ambition is to discuss molecular mechanisms as the title suggests. Significant efforts have recently been undertaken to define PAR target and target sites (e.g. PMID 29480802, 28190768, 30110646, 29954836, 30157440, 30798302, 28695509, 27256882) and there is also continuous work being performed on PAR binding proteins and their functions (reviewed for instance in PMID 23268355 and 26673700). In light of the immense interest in PARP functions for genome stability and that PARP inhibitors are mostly applied in this context, but also given the recent progress in understanding the underlying mechanisms, a more thorough discussion of recent developments would seem beneficial (even if the main focus of this review is more on DNA repair unrelated PARP functions). The authors may find inspiration in other relevant reviews (e.g. PMID 31421928, 28676700) and recent research articles (e.g. PMID 27498558, 27443740, 29950726, 29983321, 29992957, 31185216, just to list a few). Although a complete discussion may be beyond the scope of this review, mentioning and at least briefly explaining key concepts associated with PARP and PARP inhibitor functions such as synthetic lethality, PARP trapping and also replication fork stability (and reversal/degradation) in BRCA-proficient and -deficient settings (together with PARP inhibitor sensitivity and resistance) seems appropriate for any up-to-date review on the topic. The quality/resolution and size of the figures could be improved. Figure 2: Several spelling mistakes (e.g. “cardiorascular”, neurodogeratuve”), more care should be taken during the figure preparation. Figure 3: Explain better B- vs. D-NHEJ. Throughout the manuscript it would be better to refrain from writing “PARP1 inhibitor(s)” and use “PARP inhibitor(s)” instead. Introduction (especially lines 33-41 and/or Fig. 1), please clarify which family members make poly, mono, or are inactive. It may also make sense to at least briefly mention PARP inhibitors and their clinical relevance in the introduction. Line 27: As it is written, it may not be clear for a non-expert what MARylation and PARylation means. Please try to clarify, e.g. by adding “respectively” to the end of the sentence. Line 31: Better: “for a splice variant of PARP13, PARP13.2, …” Line 51: Include serines (and potentially other amino acids) as automodification sites, see comment on ADP-ribosylation target sites above and the relevant literature. Line 61: “PAR” instead of “PARs”. Line 66: “of the PARP family”. Line 107: “reported to bind”. Line 153 on HMGB1: Could more information on the underlying mechanism be provided? Line 245: “trigger PARP to synthesize” Line 253 on MRN: Could more information about the mechanism of recruitment be provided? Line 273: Check grammar. Lines 295/296: Check grammar. Lines 319-323: Does not seem to make sense to write that the mechanism how AIF breaks chromatin remains unknown and in the next sentence cite and discuss the paper that actually explains this mechanism (i.e. via MIF). Lines 340-342: Check grammar. Line 344 on Iduna: Could more information about the mechanism of PAR binding be provided? Line 352: Why plural “sequences”? Lines 419/424: Repetition “in addition”. Line 425: “helps”.
Author Response
Comments and Suggestions for Authors
Ke et al. provide an overview of PARP1 functions and on how some of them may relate to disease. The review focuses mostly on PARP1, other PARP family members are discussed in less detail, partly because they are less studied. Functional aspects that are covered include roles in regulation of chromatin, transcription and RNA metabolism, DNA repair, cellular metabolism, and PARP1-dependent cell death. The manuscript ends with a short paragraph on PARP inhibitors and their therapeutic potential. Due the growing interest in PARP inhibitors and PARP biology, many comprehensive reviews have already been published on the topic, and it may thus be difficult to find a new and original angle. Nevertheless, the authors could try to work more towards providing a synthesis of newly emerging concepts rather than listing individual and sometimes poorly connected research findings.
Reply:
We thank the reviewer for giving us the opportunity to improve our work and revise the manuscript. The comments and suggestions of the reviewer are constructive which helped us to significantly improve the quality of the manuscript. In this revised version, changes to our manuscript were all highlighted within the document by using red colored text.
Specific points:
Given that the main focus is on PARP1, the title seems too broad. Along the same line, the manuscript only scratches the surface of the roles of PARPs and PARP inhibitors in different cancers, and it would therefore be more appropriate to choose a title more specific than “Human Diseases” and perhaps focus the manuscript accordingly, e.g. on inflammation and metabolic disorders.
Reply:
As the reviewer’s suggestion, we have changed our title as to
“The Role of PARPs in Inflammation- and Metabolic- related diseases: Molecular Mechanisms and Beyond”
The manuscript would benefit from more clearly discriminating, whenever possible, between PARP functions that depend on covalent modification of target proteins with PAR versus functions that are associated with non-covalent interactions with PAR chains, especially if the ambition is to discuss molecular mechanisms as the title suggests. Significant efforts have recently been undertaken to define PAR target and target sites (e.g. PMID 29480802, 28190768, 30110646, 29954836, 30157440, 30798302, 28695509, 27256882) and there is also continuous work being performed on PAR binding proteins and their functions (reviewed for instance in PMID 23268355 and 26673700).
Reply:
We thank the reviewer and we have added this part of the content
Page1
Line 31-36
“In addition to covalent PARylating proteins on classical amino acids such as glutamic acid, aspartic acid and lysine, new amino acid sites such as arginine, cysteines, tyrosine and serine have been identified based on new experimental methods and techniques [5-11]. Notably, besides covalently PARylating target proteins, binding of proteins with PAR in noncovalent manner has also been reported to participate in a variety of biological processes [12,13] ”.
And
Page5
Line 194-201
“Interestingly, PAR generated by PARP1 and PARG also been reported participate in transcriptional regulation. For example, p53 displays a noncovalent high-affinity interaction with PAR. Non-covalent bind by free PAR and covalent PARylation of p53 orchestrates each other, affected p53-dependent DNA binding properties, transcriptional function under genotoxic stress condition [56]. Another example showed that the nuclear free PAR could be further broken down by pyrophosphatase NUDIX5 to produce ATP in the presence of pyrophosphate. The ATP then participated in the ATP-dependent chromatin remodeling and transcriptional programmes in response to hormone-dependent signaling [68] ”.
Palazzo, L.; Leidecker, O.; Prokhorova, E.; Dauben, H.; Matic, I.; Ahel, I. Serine is the major residue for ADP-ribosylation upon DNA damage. eLife 2018, 7, doi:10.7554/eLife.34334. Bonfiglio, J.J.; Fontana, P.; Zhang, Q.; Colby, T.; Gibbs-Seymour, I.; Atanassov, I.; Bartlett, E.; Zaja, R.; Ahel, I.; Matic, I. Serine ADP-Ribosylation Depends on HPF1. Molecular cell 2017, 65, 932-940 e936, doi:10.1016/j.molcel.2017.01.003. Leutert, M.; Menzel, S.; Braren, R.; Rissiek, B.; Hopp, A.K.; Nowak, K.; Bisceglie, L.; Gehrig, P.; Li, H.; Zolkiewska, A., et al. Proteomic Characterization of the Heart and Skeletal Muscle Reveals Widespread Arginine ADP-Ribosylation by the ARTC1 Ectoenzyme. Cell reports 2018, 24, 1916-1929 e1915, doi:10.1016/j.celrep.2018.07.048. Larsen, S.C.; Hendriks, I.A.; Lyon, D.; Jensen, L.J.; Nielsen, M.L. Systems-wide Analysis of Serine ADP-Ribosylation Reveals Widespread Occurrence and Site-Specific Overlap with Phosphorylation. Cell reports 2018, 24, 2493-2505 e2494, doi:10.1016/j.celrep.2018.07.083. Hendriks, I.A.; Larsen, S.C.; Nielsen, M.L. An Advanced Strategy for Comprehensive Profiling of ADP-ribosylation Sites Using Mass Spectrometry-based Proteomics. Molecular & cellular proteomics : MCP 2019, 18, 1010-1026, doi:10.1074/mcp.TIR119.001315. Larsen, S.C.; Leutert, M.; Bilan, V.; Martello, R.; Jungmichel, S.; Young, C.; Hottiger, M.O.; Nielsen, M.L. Proteome-Wide Identification of In Vivo ADP-Ribose Acceptor Sites by Liquid Chromatography-Tandem Mass Spectrometry. Methods Mol Biol 2017, 1608, 149-162, doi:10.1007/978-1-4939-6993-7_11. Gibson, B.A.; Zhang, Y.; Jiang, H.; Hussey, K.M.; Shrimp, J.H.; Lin, H.; Schwede, F.; Yu, Y.; Kraus, W.L. Chemical genetic discovery of PARP targets reveals a role for PARP-1 in transcription elongation. Science 2016, 353, 45-50, doi:10.1126/science.aaf7865. Krietsch, J.; Rouleau, M.; Pic, E.; Ethier, C.; Dawson, T.M.; Dawson, V.L.; Masson, J.Y.; Poirier, G.G.; Gagne, J.P. Reprogramming cellular events by poly(ADP-ribose)-binding proteins. Molecular aspects of medicine 2013, 34, 1066-1087, doi:10.1016/j.mam.2012.12.005. Teloni, F.; Altmeyer, M. Readers of poly(ADP-ribose): designed to be fit for purpose. Nucleic acids research 2016, 44, 993-1006, doi:10.1093/nar/gkv1383. Fischbach, A.; Kruger, A.; Hampp, S.; Assmann, G.; Rank, L.; Hufnagel, M.; Stockl, M.T.; Fischer, J.M.F.; Veith, S.; Rossatti, P., et al. The C-terminal domain of p53 orchestrates the interplay between non-covalent and covalent poly(ADP-ribosyl)ation of p53 by PARP1. Nucleic acids research 2018, 46, 804-822, doi:10.1093/nar/gkx1205. Wright, R.H.; Lioutas, A.; Le Dily, F.; Soronellas, D.; Pohl, A.; Bonet, J.; Nacht, A.S.; Samino, S.; Font-Mateu, J.; Vicent, G.P., et al. ADP-ribose-derived nuclear ATP synthesis by NUDIX5 is required for chromatin remodeling. Science 2016, 352, 1221-1225, doi:10.1126/science.aad9335.
In light of the immense interest in PARP functions for genome stability and that PARP inhibitors are mostly applied in this context, but also given the recent progress in understanding the underlying mechanisms, a more thorough discussion of recent developments would seem beneficial (even if the main focus of this review is more on DNA repair unrelated PARP functions). The authors may find inspiration in other relevant reviews (e.g. PMID 31421928, 28676700) and recent research articles (e.g. PMID 27498558, 27443740, 29950726, 29983321, 29992957, 31185216, just to list a few). Although a complete discussion may be beyond the scope of this review, mentioning and at least briefly explaining key concepts associated with PARP and PARP inhibitor functions such as synthetic lethality, PARP trapping and also replication fork stability (and reversal/degradation) in BRCA-proficient and -deficient settings (together with PARP inhibitor sensitivity and resistance) seems appropriate for any up-to-date review on the topic.
Reply:
We thank the reviewer and we have added this part of the content
Page14
Line 508-510
“To date, the use of synthetic lethality has been most effectively exploited in the context of BRCA1/2-deficient breast and ovarian cancers [161-163]”.
And
Page14
Line 531-543
“In addition, excessive PARP activation is often observed in many diseases. PARPs are key players in the inflammatory response and the inhibition of PARPs limits the proliferation of many cancer cells [26,27,95-100]. Therefore, methods targeting PARPs have potential therapeutic applications in both cancer and inflammatory diseases. Unfortunately, current research findings showed that in patients as well as in mouse models expressed PARP inhibitor resistance. This resistance is mainly due to overexpression of the drug efflux transporter gene; directly affects the activity of PARPs; leads to HR reactivation and affects the stability of the replication fork [179-186]. In addition, most of the current PARPs' inhibitors have a relatively broad-spectrum for the PARP family; however, it has become increasingly evident that the PARP family members have very different functional roles in specific physiological processes (see above). Therefore, it should be realized that the development of combination treatments of PARP inhibitor with HR-inhibiting or other drugs and specific inhibitions for PARP family members may have pleiotropic effects and pave the way for novel treatment options to target resistance mechanisms.”.
Domagala, P.; Huzarski, T.; Lubinski, J.; Gugala, K.; Domagala, W. PARP-1 expression in breast cancer including BRCA1-associated, triple negative and basal-like tumors: possible implications for PARP-1 inhibitor therapy. Breast cancer research and treatment 2011, 127, 861-869, doi:10.1007/s10549-011-1441-2. Murnyak, B.; Kouhsari, M.C.; Hershkovitch, R.; Kalman, B.; Marko-Varga, G.; Klekner, A.; Hortobagyi, T. PARP1 expression and its correlation with survival is tumour molecular subtype dependent in glioblastoma. Oncotarget 2017, 8, 46348-46362, doi:10.18632/oncotarget.18013. Beck, C.; Robert, I.; Reina-San-Martin, B.; Schreiber, V.; Dantzer, F. Poly(ADP-ribose) polymerases in double-strand break repair: focus on PARP1, PARP2 and PARP3. Experimental cell research 2014, 329, 18-25, doi:10.1016/j.yexcr.2014.07.003. Livraghi, L.; Garber, J.E. PARP inhibitors in the management of breast cancer: current data and future prospects. BMC medicine 2015, 13, 188, doi:10.1186/s12916-015-0425-1. Csonka, T.; Murnyak, B.; Szepesi, R.; Kurucz, A.; Klekner, A.; Hortobagyi, T. Poly(ADP-ribose) polymerase-1 (PARP1) and p53 labelling index correlates with tumour grade in meningiomas. Folia neuropathologica 2014, 52, 111-120. Mazzotta, A.; Partipilo, G.; De Summa, S.; Giotta, F.; Simone, G.; Mangia, A. Nuclear PARP1 expression and its prognostic significance in breast cancer patients. Tumour biology : the journal of the International Society for Oncodevelopmental Biology and Medicine 2016, 37, 6143-6153, doi:10.1007/s13277-015-4465-0. Noordermeer, S.M.; van Attikum, H. PARP Inhibitor Resistance: A Tug-of-War in BRCA-Mutated Cells. Trends in cell biology 2019, 10.1016/j.tcb.2019.07.008, doi:10.1016/j.tcb.2019.07.008. Ray Chaudhuri, A.; Nussenzweig, A. The multifaceted roles of PARP1 in DNA repair and chromatin remodelling. Nature reviews. Molecular cell biology 2017, 18, 610-621, doi:10.1038/nrm.2017.53. Ding, X.; Ray Chaudhuri, A.; Callen, E.; Pang, Y.; Biswas, K.; Klarmann, K.D.; Martin, B.K.; Burkett, S.; Cleveland, L.; Stauffer, S., et al. Synthetic viability by BRCA2 and PARP1/ARTD1 deficiencies. Nature communications 2016, 7, 12425, doi:10.1038/ncomms12425. Ray Chaudhuri, A.; Callen, E.; Ding, X.; Gogola, E.; Duarte, A.A.; Lee, J.E.; Wong, N.; Lafarga, V.; Calvo, J.A.; Panzarino, N.J., et al. Replication fork stability confers chemoresistance in BRCA-deficient cells. Nature 2016, 535, 382-387, doi:10.1038/nature18325. Maya-Mendoza, A.; Moudry, P.; Merchut-Maya, J.M.; Lee, M.; Strauss, R.; Bartek, J. High speed of fork progression induces DNA replication stress and genomic instability. Nature 2018, 559, 279-284, doi:10.1038/s41586-018-0261-5. Hanzlikova, H.; Kalasova, I.; Demin, A.A.; Pennicott, L.E.; Cihlarova, Z.; Caldecott, K.W. The Importance of Poly(ADP-Ribose) Polymerase as a Sensor of Unligated Okazaki Fragments during DNA Replication. Molecular cell 2018, 71, 319-331 e313, doi:10.1016/j.molcel.2018.06.004. Michelena, J.; Lezaja, A.; Teloni, F.; Schmid, T.; Imhof, R.; Altmeyer, M. Analysis of PARP inhibitor toxicity by multidimensional fluorescence microscopy reveals mechanisms of sensitivity and resistance. Nature communications 2018, 9, 2678, doi:10.1038/s41467-018-05031-9. Gogola, E.; Duarte, A.A.; de Ruiter, J.R.; Wiegant, W.W.; Schmid, J.A.; de Bruijn, R.; James, D.I.; Llobet, S.G.; Vis, D.J.; Annunziato, S., et al. Selective Loss of PARG Restores PARylation and Counteracts PARP Inhibitor-Mediated Synthetic Lethality. Cancer cell 2019, 35, 950-952, doi:10.1016/j.ccell.2019.05.012.
The quality/resolution and size of the figures could be improved.
Reply:
We thank the reviewer and have corrected accordingly.
Figure 2: Several spelling mistakes (e.g. “cardiorascular”, neurodogeratuve”), more care should be taken during the figure preparation.
Reply:
We thank the reviewer and have corrected accordingly.
Figure 3: Explain better B- vs. D-NHEJ.
Reply:
We thank the reviewer and we have added this part of the content
Page7
Line 290-294
“Mammalian cells employ at least two subpathways of non-homologous end-joining for the repair of DNA double strand breaks. One is the canonical DNA-PK complex (Ku70/Ku80 and DNA-PKcs)-dependent form of non-homologous end-joining (D-NHEJ); another is an alternative, slowly operating, error-prone backup pathway (B-NHEJ) [90,91]”.
Manova, V.; Singh, S.K.; Iliakis, G. Processing of DNA double strand breaks by alternative non-homologous end-joining in hyperacetylated chromatin. Genome integrity 2012, 3, 4, doi:10.1186/2041-9414-3-4. Singh, S.K.; Bednar, T.; Zhang, L.; Wu, W.; Mladenov, E.; Iliakis, G. Inhibition of B-NHEJ in plateau-phase cells is not a direct consequence of suppressed growth factor signaling. International journal of radiation oncology, biology, physics 2012, 84, e237-243, doi:10.1016/j.ijrobp.2012.03.060.
Throughout the manuscript it would be better to refrain from writing “PARP1 inhibitor(s)” and use “PARP inhibitor(s)” instead.
Reply:
We thank the reviewer and have corrected accordingly.
Introduction (especially lines 33-41 and/or Fig. 1), please clarify which family members make poly, mono, or are inactive. It may also make sense to at least briefly mention PARP inhibitors and their clinical relevance in the introduction.
Reply:
We thank the reviewer and we have added this part of the content
Page2
Line 48-51
“As Fig.1 showed, PARP1, PARP2, and PARP5 catalyze PAR attaching to target, while other PARP members (PARP 3, 4, 6-8, and 10-16) catalyze MAR to target proteins. Two members of this family, PARP 9 and 13 appear to lack any enzymatic activity”.
Line 27: As it is written, it may not be clear for a non-expert what MARylation and PARylation means. Please try to clarify, e.g. by adding “respectively” to the end of the sentence.
Line 31: Better: “for a splice variant of PARP13, PARP13.2, …”
Reply:
We thank the reviewer and have corrected accordingly.
Line 51: Include serines (and potentially other amino acids) as automodification sites, see comment on ADP-ribosylation target sites above and the relevant literature.
Reply:
We thank the reviewer and we have added this part of the content
Page1
Line 31-34
“In addition to covalent PARylating proteins on classical amino acids such as glutamic acid, aspartic acid and lysine, new amino acid sites such as arginine, cysteines, tyrosine and serine have been identified based on new experimental methods and techniques [5-11]”.
And
Page7
Line 310-318
“Recently, a list of studies has shown that Ser ADP-ribosylation (S-ADPr) is involved in genome stability regulation and the DNA damage response (DDR) processes. This modification is highly dependent on histone PARylation factor 1 (HPF1). Besides PARP1 itself , S-ADPr has been found as a widespread PTM that targets hundreds of proteins, such as histones and HMG, those proteins have important roles in chromatin regulation, DNA repair, and cancer progression, which imply the significant of S-ADPr in the regulation of DNA damage response and the maintenance of genome stability [5,6,8]. The discovery of S-ADPr have added considerable depth to our understanding of the function and versatility of ADPr signalling in the cell, and the potential physiological and pathological significance needs further study”.
Palazzo, L.; Leidecker, O.; Prokhorova, E.; Dauben, H.; Matic, I.; Ahel, I. Serine is the major residue for ADP-ribosylation upon DNA damage. eLife 2018, 7, doi:10.7554/eLife.34334. Bonfiglio, J.J.; Fontana, P.; Zhang, Q.; Colby, T.; Gibbs-Seymour, I.; Atanassov, I.; Bartlett, E.; Zaja, R.; Ahel, I.; Matic, I. Serine ADP-Ribosylation Depends on HPF1. Molecular cell 2017, 65, 932-940 e936, doi:10.1016/j.molcel.2017.01.003. Leutert, M.; Menzel, S.; Braren, R.; Rissiek, B.; Hopp, A.K.; Nowak, K.; Bisceglie, L.; Gehrig, P.; Li, H.; Zolkiewska, A., et al. Proteomic Characterization of the Heart and Skeletal Muscle Reveals Widespread Arginine ADP-Ribosylation by the ARTC1 Ectoenzyme. Cell reports 2018, 24, 1916-1929 e1915, doi:10.1016/j.celrep.2018.07.048. Larsen, S.C.; Hendriks, I.A.; Lyon, D.; Jensen, L.J.; Nielsen, M.L. Systems-wide Analysis of Serine ADP-Ribosylation Reveals Widespread Occurrence and Site-Specific Overlap with Phosphorylation. Cell reports 2018, 24, 2493-2505 e2494, doi:10.1016/j.celrep.2018.07.083. Hendriks, I.A.; Larsen, S.C.; Nielsen, M.L. An Advanced Strategy for Comprehensive Profiling of ADP-ribosylation Sites Using Mass Spectrometry-based Proteomics. Molecular & cellular proteomics : MCP 2019, 18, 1010-1026, doi:10.1074/mcp.TIR119.001315. Larsen, S.C.; Leutert, M.; Bilan, V.; Martello, R.; Jungmichel, S.; Young, C.; Hottiger, M.O.; Nielsen, M.L. Proteome-Wide Identification of In Vivo ADP-Ribose Acceptor Sites by Liquid Chromatography-Tandem Mass Spectrometry. Methods Mol Biol 2017, 1608, 149-162, doi:10.1007/978-1-4939-6993-7_11. Gibson, B.A.; Zhang, Y.; Jiang, H.; Hussey, K.M.; Shrimp, J.H.; Lin, H.; Schwede, F.; Yu, Y.; Kraus, W.L. Chemical genetic discovery of PARP targets reveals a role for PARP-1 in transcription elongation. Science 2016, 353, 45-50, doi:10.1126/science.aaf7865.
Line 61: “PAR” instead of “PARs”.
Line 66: “of the PARP family”.
Line 107: “reported to bind”.
Reply:
We thank the reviewer and have corrected accordingly.
Line 153 on HMGB1: Could more information on the underlying mechanism be provided?
Reply:
We thank the reviewer and we have added this part of the content
Page5
Line 183-188
“The translocation of HMGB1 from the nucleus to the cytoplasm is dependent on the modification of HMGB1 itself. Its reported acetylation of HMGB1 is necessary for HMGB1 shuttling and exported acetyl-HMGB1 cannot return to the nucleus. Studies showed that PARP1 catalyzed PARylation of HMGB1, which facilitated acetylation of HMGB1. The crosstalk between PARylation and acetylation directed the migration of HMGB1 into the cytoplasm as well as release into the extracellular space [64,65]”.
Yang, Z.; Li, L.; Chen, L.; Yuan, W.; Dong, L.; Zhang, Y.; Wu, H.; Wang, C. PARP-1 mediates LPS-induced HMGB1 release by macrophages through regulation of HMGB1 acetylation. J Immunol 2014, 193, 6114-6123, doi:10.4049/jimmunol.1400359. Bonaldi, T.; Talamo, F.; Scaffidi, P.; Ferrera, D.; Porto, A.; Bachi, A.; Rubartelli, A.; Agresti, A.; Bianchi, M.E. Monocytic cells hyperacetylate chromatin protein HMGB1 to redirect it towards secretion. The EMBO journal 2003, 22, 5551-5560, doi:10.1093/emboj/cdg516.
Line 245: “trigger PARP to synthesize”
Reply:
We thank the reviewer and have corrected accordingly.
Line 253 on MRN: Could more information about the mechanism of recruitment be provided?
Reply:
We thank the reviewer and we have added this part of the content
Page7
Line 296-301
“Then the activated PARP1 rapidly recruits the DNA damage sensors meiotic recombination 11 (Mre11) and Nijmegen breakage syndrome protein 1 (Nbs1) to sites of DSBs, Mre11 has a putative PAR binding domain, suggesting that its recruitment could be dependent on the activity of PARP1. Mre11 protein interacts with the Rad50 recombinase to form MRN (Mre11-Rad50-Nbs1) complex, after which the MRN complex initiates the repair process [88,92]”.
Jeggo, P.A.; Pearl, L.H.; Carr, A.M. DNA repair, genome stability and cancer: a historical perspective. Nature reviews. Cancer 2016, 16, 35-42, doi:10.1038/nrc.2015.4. Haince, J.F.; McDonald, D.; Rodrigue, A.; Dery, U.; Masson, J.Y.; Hendzel, M.J.; Poirier, G.G. PARP1-dependent kinetics of recruitment of MRE11 and NBS1 proteins to multiple DNA damage sites. The Journal of biological chemistry 2008, 283, 1197-1208, doi:10.1074/jbc.M706734200.
Line 273: Check grammar.
Lines 295/296: Check grammar.
Reply:
We thank the reviewer and have corrected accordingly.
Lines 319-323: Does not seem to make sense to write that the mechanism how AIF breaks chromatin remains unknown and in the next sentence cite and discuss the paper that actually explains this mechanism (i.e. via MIF).
Lines 340-342: Check grammar.
Reply:
We thank the reviewer and have corrected accordingly.
Line 344 on Iduna: Could more information about the mechanism of PAR binding be provided?
Reply:
We thank the reviewer and we have added this part of the content
Page11
Line 397-403
“Iduna, an NMDA-induced neuroprotective protein, has been reported to protects against stroke induced neuronal injury. Iduna contains a PBM domain consists approximately 20 amino acids, PBM domain is high affinity with PAR, by this way, Iduna competed with AIF for PAR binding, thereby blocking the translocation of AIF from mitochondria to the nucleus. Moreover, Iduna also been found is a PAR-dependent E3 ligase that binds and ubiquitinates both PARylated and PAR binding proteins via its PBM domain, marking these proteins for ubiquitin proteasomal degradation [96,121]”.
Murnyak, B.; Kouhsari, M.C.; Hershkovitch, R.; Kalman, B.; Marko-Varga, G.; Klekner, A.; Hortobagyi, T. PARP1 expression and its correlation with survival is tumour molecular subtype dependent in glioblastoma. Oncotarget 2017, 8, 46348-46362, doi:10.18632/oncotarget.18013. Kang, H.C.; Lee, Y.I.; Shin, J.H.; Andrabi, S.A.; Chi, Z.; Gagne, J.P.; Lee, Y.; Ko, H.S.; Lee, B.D.; Poirier, G.G., et al. Iduna is a poly(ADP-ribose) (PAR)-dependent E3 ubiquitin ligase that regulates DNA damage. Proceedings of the National Academy of Sciences of the United States of America 2011, 108, 14103-14108, doi:10.1073/pnas.1108799108.
Line 352: Why plural “sequences”?
Lines 419/424: Repetition “in addition”.
Line 425: “helps”.
Reply:
We thank the reviewer and have corrected accordingly.

Reviewer 2 Report
he manuscript "The role of PARPs in human diseases: molecular mechanisms and beyond", written by Ke Y, Wang C, Zhang J, Zhong X, Wang R, Zeng X and Ba X is a review describing different roles of PARP enzymes in cancer and diseases.
The Introduction describes molecular structure of different members of the PARP superfamily. The second section presents the role of PARP1 in chromatin modulation, transcriptional and RNA regulation through numerous examples of PARP1 involvement in gene regulation and protein expression in conditions which could be connected with diseases. I would suggest to explain that PARP1 can influence other molecules through protein-protein interactions or through PARyation and to describe under which circumstances PARP1 acts, whether there is its activation induced by DNA damage or kinases, for each example. Many of the conditions described connect PARP1 with some kind of inflammatory response and more details on the overall circumstances would better explain the role of PARP .
In the third section, the role of PARP1 in DNA damage repair is described. There is no explanation why should PARP expression be elevated in cancer tissues, and what does it mean that it is a good candidate for sensitizing cancer cells to the cytotoxic effects of DNA damaging agents. Also, if the PARP activity is elevated in cancer cells, it could be explained why, with more references.
The fourth section describes PARP-dependent death and related diseases. AIF-dependent parthanatos is described in details. Parthanatos is involved in several neurodegenerative disorders such as Parkinson disease and stroke. The role of PARP in the stroke should be described in more details, considering excitotoxicity and specific ways of PARP activation. On the other side, the role of estrogen receptors inhibiting PARP activation should be described separately, as these experiments were done on breast cancer cells.
In the fifth section the roles of other PARPs in human diseases are described. Describing the role of PARP10, the authors continue to describe the role of PARP1 in NFkappaB activation. This part should be incorporated in the above section describing the role of PARP1 in inflammation. Beside that, interactions of PARP1 with p50 lead to stabilization of p50 interactions and activation of NFkappaB dependent transcription (not NFkappaB activation). As above, it would be more clear to explain the circumstances of the PARP9, 14 etc. activities, and to put them in the context of inflammation or the presence of cytokines. On the other side, the role of PARP16 in the response on the endoplasmic reticulum stress should have a separate paragraph.
The sixth section describes therapeutic approaches for targeting PARPs. It should be stressed that nowadays the rationale for use PARP inhibitors in cancer therapy is so called synthetic lethality and is used for tumors deficient in some types of DNA damage repair. It is used in combination with DNA damaging agents. As BRCA mutations can be associated with breast cancer, PARP inhibitors are also used for their treatment.
In Conclusion, I think, citing different inflammatory diseases in brackets can be omitted. Instead, the role of PARP in them can be described above (such as roles in diabetes, atherosclerosis etc.). In the manuscript, PARP role in the diseases was analyzed from the point of its activities and types, and additional section could be added with description of the PARP role from the point of view of different diseases, whether it is involved in the process of inflammation which could be in the etiology of the particular disease, in apoptosis or disturbed process of differentiation.
Minor
row 119: effector
146: NFAT
211: cytoplasmic HuR expression is elevated – better cytoplasmic localization
Figure2: a) in type III EBV latency
251: there is no explanation what is B-NHEJ
302: intuitive experimental data: unclear expression
305 AIF in mitochondria – from mitochondria
308: mild DNA breaks – low level of DNA damage
319: AIF breaks chromatin stability: change the expression
Figure 3: 328: induction of the cell death
338: "both impact PARP activation, thereby haltering cell death" – they influence (how) PARP activation, and increase cell viability
341: "Our experimental results suggest that hormonal therapy in the intervention of PARP1 activation-associated diseases." sentence unfinished
425: PARP13 also helps...
440: ...implicating that PARP14 is an important....
442: roles to PARP14...
458: the most widely characterized for these tumors – change the expression
459: lacking DNA repair defects - change the expression
478: PARPs'
Author Response
Comments and Suggestions for Authors
the manuscript "The role of PARPs in human diseases: molecular mechanisms and beyond", written by Ke Y, Wang C, Zhang J, Zhong X, Wang R, Zeng X and Ba X is a review describing different roles of PARP enzymes in cancer and diseases.
Reply:
We appreciate Reviewer’ comments about our work, and we thank you for giving us the opportunity to reply and resubmit our revised manuscript for possible publication in cells. In this revised version, changes to our manuscript were all highlighted within the document by using red colored text.
The Introduction describes molecular structure of different members of the PARP superfamily. The second section presents the role of PARP1 in chromatin modulation, transcriptional and RNA regulation through numerous examples of PARP1 involvement in gene regulation and protein expression in conditions which could be connected with diseases. I would suggest to explain that PARP1 can influence other molecules through protein-protein interactions or through PARyation and to describe under which circumstances PARP1 acts, whether there is its activation induced by DNA damage or kinases, for each example. Many of the conditions described connect PARP1 with some kind of inflammatory response and more details on the overall circumstances would better explain the role of PARP .
Reply:
We thank the reviewer and we have added this part of the content
Page2-3
Line 73-84
“PARP1 is usually activated by DNA damage, and the modified target is primarily PARP1 itself, which is called PARP1 automodification. Furthermore, binding to non-damaged DNA structures is considered to be a valid determinant of PARP1 activation in the absence of abundant DNA fragmentation [19]. And besides direct binding to DNA, PARP1 activity can also be stimulated by interactions with protein binding partners. For example, the DNA binding transcription factor Yin Yang 1 (YY1) could bind to the BRCT motif in the automodification domain of PARP1, then stimulate PARP1 enzymatic activity [20]. In addition, studies have shown that post-translational modifications such as phosphorylation or acetylation, etc. are also alternative mechanisms of PARP1 activity [21]. Recently, our study showed that in response to LPS or TNF-α exposure, the non-receptor tyrosine kinase c-Abl phosphorylates PARP1 at the conserved Y829 site. Tyrosine phosphorylation of PARP1 is required for PARP1 catalytic activity and NF-κB-dependent proinflammatory gene regulation [22]. Actually, a large body of data has now shown that PARP1 functions as a cellular rheostat, promoting different cellular response upon a wide range of type, duration, and strength of stress signals [4]. As the strength of stress stimulus increases, the levels of PARP1 activity and PAR synthesis increase, leading to different cellular outcomes. Weakly activity of PARP1 caused by LPS or TNFα can help cells response inflammation [22-24]; Mild or moderate stresses leads to transcription and DNA repair responses that help to maintain genome stability, in contrast, excessive activity of PARP1 resulted the depletion of NAD+ and accumulation of PAR, leading some metabolic disorder or cell death [4].”.
Lonskaya, I.; Potaman, V.N.; Shlyakhtenko, L.S.; Oussatcheva, E.A.; Lyubchenko, Y.L.; Soldatenkov, V.A. Regulation of poly(ADP-ribose) polymerase-1 by DNA structure-specific binding. The Journal of biological chemistry 2005, 280, 17076-17083, doi:M413483200 [pii]
10.1074/jbc.M413483200.
Doetsch, M.; Gluch, A.; Poznanovic, G.; Bode, J.; Vidakovic, M. YY1-binding sites provide central switch functions in the PARP-1 gene expression network. PloS one 2012, 7, e44125, doi:10.1371/journal.pone.0044125. Piao, L.; Fujioka, K.; Nakakido, M.; Hamamoto, R. Regulation of poly(ADP-Ribose) polymerase 1 functions by post-translational modifications. Front Biosci (Landmark Ed) 2018, 23, 13-26. Bohio, A.A.; Sattout, A.; Wang, R.; Wang, K.; Sah, R.K.; Guo, X.; Zeng, X.; Ke, Y.; Boldogh, I.; Ba, X. c-Abl-Mediated Tyrosine Phosphorylation of PARP1 Is Crucial for Expression of Proinflammatory Genes. J Immunol 2019, 10.4049/jimmunol.1801616, doi:10.4049/jimmunol.1801616. Liu, L.; Ke, Y.; Jiang, X.; He, F.; Pan, L.; Xu, L.; Zeng, X.; Ba, X. Lipopolysaccharide activates ERK-PARP-1-RelA pathway and promotes nuclear factor-kappaB transcription in murine macrophages. Hum Immunol 2012, 73, 439-447, doi:10.1016/j.humimm.2012.02.002. Ke, Y.; Han, Y.; Guo, X.; Wen, J.; Wang, K.; Jiang, X.; Tian, X.; Ba, X.; Boldogh, I.; Zeng, X. Erratum: PARP1 promotes gene expression at the post-transcriptional level by modulating the RNA-binding protein HuR. Nature communications 2017, 8, 15191, doi:10.1038/ncomms15191.
In the third section, the role of PARP1 in DNA damage repair is described. There is no explanation why should PARP expression be elevated in cancer tissues, and what does it mean that it is a good candidate for sensitizing cancer cells to the cytotoxic effects of DNA damaging agents. Also, if the PARP activity is elevated in cancer cells, it could be explained why, with more references.
Reply:
We thank the reviewer and we have added this part of the content
Page7
Line 304-306
“PARP1 is overexpressed in a variety of cancers, including glioblastoma, prostate and breast cancer [95-100]. Up-regulated PARP1 enhances the anti-apoptotic properties of tumors, leading to resistance to DNA damage therapeutics. Cancer cells use PARP1 to initiate a variety of DNA repair pathways, particularly BER and HR, to prevent cell death caused by the accumulation of cytotoxic DNA damage”.
Domagala, P.; Huzarski, T.; Lubinski, J.; Gugala, K.; Domagala, W. PARP-1 expression in breast cancer including BRCA1-associated, triple negative and basal-like tumors: possible implications for PARP-1 inhibitor therapy. Breast cancer research and treatment 2011, 127, 861-869, doi:10.1007/s10549-011-1441-2. Murnyak, B.; Kouhsari, M.C.; Hershkovitch, R.; Kalman, B.; Marko-Varga, G.; Klekner, A.; Hortobagyi, T. PARP1 expression and its correlation with survival is tumour molecular subtype dependent in glioblastoma. Oncotarget 2017, 8, 46348-46362, doi:10.18632/oncotarget.18013. Beck, C.; Robert, I.; Reina-San-Martin, B.; Schreiber, V.; Dantzer, F. Poly(ADP-ribose) polymerases in double-strand break repair: focus on PARP1, PARP2 and PARP3. Experimental cell research 2014, 329, 18-25, doi:10.1016/j.yexcr.2014.07.003. Livraghi, L.; Garber, J.E. PARP inhibitors in the management of breast cancer: current data and future prospects. BMC medicine 2015, 13, 188, doi:10.1186/s12916-015-0425-1. Csonka, T.; Murnyak, B.; Szepesi, R.; Kurucz, A.; Klekner, A.; Hortobagyi, T. Poly(ADP-ribose) polymerase-1 (PARP1) and p53 labelling index correlates with tumour grade in meningiomas. Folia neuropathologica 2014, 52, 111-120. Mazzotta, A.; Partipilo, G.; De Summa, S.; Giotta, F.; Simone, G.; Mangia, A. Nuclear PARP1 expression and its prognostic significance in breast cancer patients. Tumour biology : the journal of the International Society for Oncodevelopmental Biology and Medicine 2016, 37, 6143-6153, doi:10.1007/s13277-015-4465-0.
The fourth section describes PARP-dependent death and related diseases. AIF-dependent parthanatos is described in details. Parthanatos is involved in several neurodegenerative disorders such as Parkinson disease and stroke. The role of PARP in the stroke should be described in more details, considering excitotoxicity and specific ways of PARP activation. On the other side, the role of estrogen receptors inhibiting PARP activation should be described separately, as these experiments were done on breast cancer cells.
Reply:
We thank the reviewer and we have added this part of the content
Page11
Line 393-403
“Glutamate is the main excitatory neurotransmitter that regulates the normal physiological activities of the brain. Excess glutamate acts on N-methyl-d-aspartate (NMDA) receptors to induce many neurological diseases, including trauma and stroke. Glutamate excitotoxicity is mainly mediated through the influx of calcium through the NMDA receptor, leading to PARP1 activation and PAR polymer production. Iduna, an NMDA-induced neuroprotective protein, has been reported to protects against stroke induced neuronal injury. Iduna contains a PBM domain consists approximately 20 amino acids, PBM domain is high affinity with PAR, by this way, Iduna competed with AIF for PAR binding, thereby blocking the translocation of AIF from mitochondria to the nucleus. Moreover, Iduna also been found is a PAR-dependent E3 ligase that binds and ubiquitinates both PARylated and PAR binding proteins via its PBM domain, marking these proteins for ubiquitin proteasomal degradation [96,121]”.
Murnyak, B.; Kouhsari, M.C.; Hershkovitch, R.; Kalman, B.; Marko-Varga, G.; Klekner, A.; Hortobagyi, T. PARP1 expression and its correlation with survival is tumour molecular subtype dependent in glioblastoma. Oncotarget 2017, 8, 46348-46362, doi:10.18632/oncotarget.18013. Kang, H.C.; Lee, Y.I.; Shin, J.H.; Andrabi, S.A.; Chi, Z.; Gagne, J.P.; Lee, Y.; Ko, H.S.; Lee, B.D.; Poirier, G.G., et al. Iduna is a poly(ADP-ribose) (PAR)-dependent E3 ubiquitin ligase that regulates DNA damage. Proceedings of the National Academy of Sciences of the United States of America 2011, 108, 14103-14108, doi:10.1073/pnas.1108799108.
In the fifth section the roles of other PARPs in human diseases are described. Describing the role of PARP10, the authors continue to describe the role of PARP1 in NFkappaB activation. This part should be incorporated in the above section describing the role of PARP1 in inflammation. Beside that, interactions of PARP1 with p50 lead to stabilization of p50 interactions and activation of NFkappaB dependent transcription (not NFkappaB activation). As above, it would be more clear to explain the circumstances of the PARP9, 14 etc. activities, and to put them in the context of inflammation or the presence of cytokines. On the other side, the role of PARP16 in the response on the endoplasmic reticulum stress should have a separate paragraph.
Reply:
We thank the reviewer and have corrected accordingly. Since the influence of PARP1 on NF- kappaB has been extensively studied, we will not make a statement here.
Furthermore, We explored the role of PARP9 and PARP14 in IFNγ stimulation.
Page13
Line 493-498
“Another member of the PARP family, PARP9, has been reported to play opposing roles to PARP14 in macrophage activation. The knockdown of PARP14 increased proinflammatory gene expression and STAT1 phosphorylation, whereas PARP9 silencing inhibited proinflammatory gene and STAT1 phosphorylation in IFNγ-treated cells. PARP14-induced MARylation of STAT1 was found to be inhibited by PARP9 [159]”.
In addition, we separately describe the role of PARP16 in endoplasmic reticulum stress response.
Page13
Line 499-502
“Moreover, PARPs also regulate the unfolded protein response (UPR) of the endoplasmic reticulum (ER). PARP16 (also known as ARTD15) is a tail-anchored ER protein, which is inserted into the ER membrane. It appears that both PARP16 itself and its catalytic activity are required for the ER stress responses by regulating the UPR signaling pathway [160]”.
The sixth section describes therapeutic approaches for targeting PARPs. It should be stressed that nowadays the rationale for use PARP inhibitors in cancer therapy is so called synthetic lethality and is used for tumors deficient in some types of DNA damage repair. It is used in combination with DNA damaging agents. As BRCA mutations can be associated with breast cancer, PARP inhibitors are also used for their treatment.
Reply:
We thank the reviewer and we have added this part of the content
Page14
Line 508-510
“To date, the use of synthetic lethality has been most effectively exploited in the context of BRCA1/2-deficient breast and ovarian cancers [161-163]”.
And
Page14
Line 531-543
“In addition, excessive PARP activation is often observed in many diseases. PARPs are key players in the inflammatory response and the inhibition of PARPs limits the proliferation of many cancer cells [26,27,95-100]. Therefore, methods targeting PARPs have potential therapeutic applications in both cancer and inflammatory diseases. Unfortunately, current research findings showed that in patients as well as in mouse models expressed PARP inhibitor resistance. This resistance is mainly due to overexpression of the drug efflux transporter gene; directly affects the activity of PARPs; leads to HR reactivation and affects the stability of the replication fork [179-186]. In addition, most of the current PARPs' inhibitors have a relatively broad-spectrum for the PARP family; however, it has become increasingly evident that the PARP family members have very different functional roles in specific physiological processes (see above). Therefore, it should be realized that the development of combination treatments of PARP inhibitor with HR-inhibiting or other drugs and specific inhibitions for PARP family members may have pleiotropic effects and pave the way for novel treatment options to target resistance mechanisms”.
Domagala, P.; Huzarski, T.; Lubinski, J.; Gugala, K.; Domagala, W. PARP-1 expression in breast cancer including BRCA1-associated, triple negative and basal-like tumors: possible implications for PARP-1 inhibitor therapy. Breast cancer research and treatment 2011, 127, 861-869, doi:10.1007/s10549-011-1441-2. Murnyak, B.; Kouhsari, M.C.; Hershkovitch, R.; Kalman, B.; Marko-Varga, G.; Klekner, A.; Hortobagyi, T. PARP1 expression and its correlation with survival is tumour molecular subtype dependent in glioblastoma. Oncotarget 2017, 8, 46348-46362, doi:10.18632/oncotarget.18013. Beck, C.; Robert, I.; Reina-San-Martin, B.; Schreiber, V.; Dantzer, F. Poly(ADP-ribose) polymerases in double-strand break repair: focus on PARP1, PARP2 and PARP3. Experimental cell research 2014, 329, 18-25, doi:10.1016/j.yexcr.2014.07.003. Livraghi, L.; Garber, J.E. PARP inhibitors in the management of breast cancer: current data and future prospects. BMC medicine 2015, 13, 188, doi:10.1186/s12916-015-0425-1. Csonka, T.; Murnyak, B.; Szepesi, R.; Kurucz, A.; Klekner, A.; Hortobagyi, T. Poly(ADP-ribose) polymerase-1 (PARP1) and p53 labelling index correlates with tumour grade in meningiomas. Folia neuropathologica 2014, 52, 111-120. Mazzotta, A.; Partipilo, G.; De Summa, S.; Giotta, F.; Simone, G.; Mangia, A. Nuclear PARP1 expression and its prognostic significance in breast cancer patients. Tumour biology : the journal of the International Society for Oncodevelopmental Biology and Medicine 2016, 37, 6143-6153, doi:10.1007/s13277-015-4465-0. Noordermeer, S.M.; van Attikum, H. PARP Inhibitor Resistance: A Tug-of-War in BRCA-Mutated Cells. Trends in cell biology 2019, 10.1016/j.tcb.2019.07.008, doi:10.1016/j.tcb.2019.07.008. Ray Chaudhuri, A.; Nussenzweig, A. The multifaceted roles of PARP1 in DNA repair and chromatin remodelling. Nature reviews. Molecular cell biology 2017, 18, 610-621, doi:10.1038/nrm.2017.53. Ding, X.; Ray Chaudhuri, A.; Callen, E.; Pang, Y.; Biswas, K.; Klarmann, K.D.; Martin, B.K.; Burkett, S.; Cleveland, L.; Stauffer, S., et al. Synthetic viability by BRCA2 and PARP1/ARTD1 deficiencies. Nature communications 2016, 7, 12425, doi:10.1038/ncomms12425. Ray Chaudhuri, A.; Callen, E.; Ding, X.; Gogola, E.; Duarte, A.A.; Lee, J.E.; Wong, N.; Lafarga, V.; Calvo, J.A.; Panzarino, N.J., et al. Replication fork stability confers chemoresistance in BRCA-deficient cells. Nature 2016, 535, 382-387, doi:10.1038/nature18325. Maya-Mendoza, A.; Moudry, P.; Merchut-Maya, J.M.; Lee, M.; Strauss, R.; Bartek, J. High speed of fork progression induces DNA replication stress and genomic instability. Nature 2018, 559, 279-284, doi:10.1038/s41586-018-0261-5. Hanzlikova, H.; Kalasova, I.; Demin, A.A.; Pennicott, L.E.; Cihlarova, Z.; Caldecott, K.W. The Importance of Poly(ADP-Ribose) Polymerase as a Sensor of Unligated Okazaki Fragments during DNA Replication. Molecular cell 2018, 71, 319-331 e313, doi:10.1016/j.molcel.2018.06.004. Michelena, J.; Lezaja, A.; Teloni, F.; Schmid, T.; Imhof, R.; Altmeyer, M. Analysis of PARP inhibitor toxicity by multidimensional fluorescence microscopy reveals mechanisms of sensitivity and resistance. Nature communications 2018, 9, 2678, doi:10.1038/s41467-018-05031-9. Gogola, E.; Duarte, A.A.; de Ruiter, J.R.; Wiegant, W.W.; Schmid, J.A.; de Bruijn, R.; James, D.I.; Llobet, S.G.; Vis, D.J.; Annunziato, S., et al. Selective Loss of PARG Restores PARylation and Counteracts PARP Inhibitor-Mediated Synthetic Lethality. Cancer cell 2019, 35, 950-952, doi:10.1016/j.ccell.2019.05.012.
In Conclusion, I think, citing different inflammatory diseases in brackets can be omitted. Instead, the role of PARP in them can be described above (such as roles in diabetes, atherosclerosis etc.). In the manuscript, PARP role in the diseases was analyzed from the point of its activities and types, and additional section could be added with description of the PARP role from the point of view of different diseases, whether it is involved in the process of inflammation which could be in the etiology of the particular disease, in apoptosis or disturbed process of differentiation.
Reply:
We thank the reviewer and have corrected as shown below:
Line 552-554
“In conclusion, PARP controls many of the genes involved in regulating immune responses, cell survival, and inflammation and metabolic disorders. Therefore, targeting PARP has potential therapeutic applications in a number of diseases associated with inflammation and metabolism”.
Furthermore
Since the manuscript mainly focus some inflammation- and metabolic- diseases, we have changed our title as below:
“The Role of PARPs in Inflammation- and Metabolic- related diseases: Molecular Mechanisms and Beyond”
And as the reviewer’s suggestion, In addition to exploring the role of PARP1 in melanoma, metabolic disorders and pulmonary inflammation (see “2. PARP1 is implicated in disease via altering gene expression” part ) ,we have further explored the role of PARP in some specific diseases, such as:
Page11
Line 393-397
“Glutamate is the main excitatory neurotransmitter that regulates the normal physiological activities of the brain. Excess glutamate acts on N-methyl-d-aspartate (NMDA) receptors to induce many neurological diseases, including trauma and stroke. Glutamate excitotoxicity is mainly mediated through the influx of calcium through the NMDA receptor, leading to PARP1 activation and PAR polymer production”.
Minor
119: effector
146: NFAT
211: cytoplasmic HuR expression is elevated – better cytoplasmic localization
Figure2: a) in type III EBV latency
Reply:
We thank the reviewer and have corrected accordingly.
251: there is no explanation what is B-NHEJ
Reply:
We thank the reviewer and we have added this part of the content
Page7
Line 290-294
“Mammalian cells employ at least two subpathways of non-homologous end-joining for the repair of DNA double strand breaks. One is the canonical DNA-PK complex (Ku70/Ku80 and DNA-PKcs)-dependent form of non-homologous end-joining (D-NHEJ); another is an alternative, slowly operating, error-prone backup pathway (B-NHEJ) [90,91]”.
Manova, V.; Singh, S.K.; Iliakis, G. Processing of DNA double strand breaks by alternative non-homologous end-joining in hyperacetylated chromatin. Genome integrity 2012, 3, 4, doi:10.1186/2041-9414-3-4. Singh, S.K.; Bednar, T.; Zhang, L.; Wu, W.; Mladenov, E.; Iliakis, G. Inhibition of B-NHEJ in plateau-phase cells is not a direct consequence of suppressed growth factor signaling. International journal of radiation oncology, biology, physics 2012, 84, e237-243, doi:10.1016/j.ijrobp.2012.03.060.
302: intuitive experimental data: unclear expression
305 AIF in mitochondria – from mitochondria
308: mild DNA breaks – low level of DNA damage
319: AIF breaks chromatin stability: change the expression
Figure 3: 328: induction of the cell death
338: "both impact PARP activation, thereby haltering cell death" – they influence (how) PARP activation, and increase cell viability
341: "Our experimental results suggest that hormonal therapy in the intervention of PARP1 activation-associated diseases." sentence unfinished
425: PARP13 also helps...
440: ...implicating that PARP14 is an important....
442: roles to PARP14...
458: the most widely characterized for these tumors – change the expression
459: lacking DNA repair defects - change the expression
478: PARPs'
Reply:
We thank the reviewer and have corrected accordingly.
